# Increased incompatibility of heterologous algal symbionts under thermal stress in the cnidarian-dinoflagellate model Aiptasia

Maha J. Cziesielski[1,2], Yi Jin Liew[1,2,3], Guoxin Cui [1,2] & Manuel Aranda [1,2✉]

Rising ocean temperatures are increasing the rate and intensity of coral mass bleaching events, leading to the collapse of coral reef ecosystems. To better understand the dynamics of coral-algae symbioses, it is critical to decipher the role each partner plays in the holobiont's thermotolerance. Here, we investigated the role of the symbiont by comparing transcriptional heat stress responses of anemones from two thermally distinct locations, Florida (CC7) and Hawaii (H2) as well as a heterologous host-symbiont combination composed of CC7 host anemones inoculated with the symbiont *Breviolum minutum* (SSB01) from H2 anemones (CC7-B01). We find that oxidative stress and apoptosis responses are strongly influenced by symbiont type, as further confirmed by caspase-3 activation assays, but that the overall response to heat stress is dictated by the compatibility of both partners. Expression of genes essential to symbiosis revealed a shift from a nitrogen- to a carbon-limited state only in the heterologous combination CC7-B01, suggesting a bioenergetic disruption of symbiosis during stress. Our results indicate that symbiosis is highly fine-tuned towards particular partner combinations and that heterologous host-symbiont combinations are metabolically less compatible under stress. These results are essential for future strategies aiming at increasing coral resilience using heterologous thermotolerant symbionts.

[1] Marine Science Program, Biological and Environmental Sciences and Engineering Division, King Abdullah University of Science and Technology (KAUST), Thuwal 23955-6900, Kingdom of Saudi Arabia. [2] Red Sea Research Center, King Abdullah University of Science and Technology (KAUST), Thuwal 23955-6900, Kingdom of Saudi Arabia. [3] Present address: CSIRO Health & Biosecurity, North Ryde, NSW, Australia. ✉email: manuel.aranda@kaust.edu.sa

Coral reefs are marine-biodiversity hotspots of significant ecological and economic importance, sustaining around 25% of all marine species[1]. Coral growth and successful reef formation in oligotrophic waters depend on the endosymbiosis between the cnidarian animal and the dinoflagellate algae of the family Symbiodiniaceae[2]. This cnidarian-algae symbiosis is an evolutionarily conserved relationship that depends on the metabolic exchange between the two partners[3]. Symbiodiniaceae, living in the gastrodermal cells of the cnidarian host, typically supply up to 90% of the hosts' energy requirement[4,5]. Although the relationship between the two partners is based on a finely tuned exchange of nutrients, overall knowledge on cellular nutrient quota and uptake are lacking[6–8] and the mechanisms underlying symbiosis establishment, maintenance, and breakdown are not fully understood[9]. Research focusing on this metabolic coupling has shown that the translocation of photosynthates, particularly energetically rich carbon compounds, from the symbiont to the host, plays a crucial role in their symbiotic interactions[10,11]. Since nitrogen availability can control Symbiodiniaceae proliferation, and hence carbon translocation to the host[12], the nutrient cycling mechanisms and translocation of nitrogen are important in stabilizing the symbiotic relationship[13–15].

The symbiotic relationship is, however, fragile and easily disturbed by environmental changes, such as increasing sea surface temperatures. Both symbionts and corals experience stress during thermal anomalies: increased temperatures lead to an excess production of reactive oxygen species (ROS) that affect the stability of the relationship between host and symbiont[16]. Additionally, and maybe more importantly, photoinhibition in the symbionts may lead to reduced carbon translocation and affect overall nutrient exchange in the symbiosis[17]. This altered symbiotic nutrient cycling may ultimately undermine the functioning of symbiosis, contributing to the breakdown of symbiosis and, subsequently, the death of the coral host[8,15].

Hope for coral survival comes from observed variations in thermal tolerance both within and between corals as well as symbiont species[18]. Due to the intricacies of their coexistence, coral thermal tolerance depends on the physiological capabilities of both partners[19,20] and the associated microbiome[21,22]. Notably, temperature-adapted symbiont species can reduce the overall stress exerted on the host and hence, increase bleaching resilience[23–26]. The different host-symbiont combinations, as well as a coral's shift in populations of associated symbionts in response to temperature stress, have further highlighted the significance of the symbionts in thermal resilience and how highly fine-tuned the symbiotic relationship is[26–30], although the degree and pervasiveness of flexibility remain uncertain[31,32]. This potentially crucial role of the symbionts in determining holobiont thermal resilience has nurtured the hypothesis that inoculating corals with more thermally tolerant symbiont species could increase the coral's resilience and mitigate the effects of climate change[33–35].

However, inoculation with heterologous thermotolerant symbionts does not necessarily improve temperature resilience of the host. Part of this may be due to the mechanisms underlying co-evolutionary processes and resulting host-symbiont specificity[36]. Many cnidarian hosts can only associate with specific symbiont species[37]. Importantly, the specificity and selectivity of the symbiotic relationship highlight the conserved evolutionary link between the two partners[38]. This raises two important points: not all cnidarian-algae combinations are successful, and symbioses established under optimal conditions may not be viable under stressful conditions. Thus, temperature resilience of the symbiont is not the only factor driving coral thermal resilience, and the role of the host in this process should not be underestimated[30,39,40]. Understanding who and what drives thermal resilience, and how

to best preserve reef integrity, is therefore inherently more complex than previously anticipated.

This study aimed to decipher how the host response to heat stress may be affected by the associated symbiont strain. To do this, we used the sea anemone Aiptasia (*Exaiptasia diaphana*), which has emerged as a flexible model system to study cnidarian-algae symbioses[41–43]. Here, we used two genotypes originating from geographically distinct locations, Florida (CC7) and Hawaii (H2), that associate with different species of Symbiodiniaceae, namely *Symbiodinium linucheae* (SSA01) and *Breviolum minutum* (SSB01) respectively[2]. Aiptasia can also be maintained aposymbiotically, allowing the reinfection of different host genotypes with different symbiont species. Thus, to further understand the role of the symbiont and symbiosis mechanism responses under stress, we manipulated host-symbiont combinations by creating a strain that consisted of the CC7 host genotype and symbiont strain SSB01 isolated from H2 anemones, referred to as CC7-B01. We refer to this strain as heterologous in this study, reflecting the fact that while it may be natural for Aiptasia to host SSB01 symbionts, the genotype CC7 is not commonly associated with this symbiont strain. As such, we refer to CC7-B01 as a heterologous combination to reflect a non-native symbiosis, while CC7 and H2 reflect the geographically native relationships and are therefore here considered as homologous. We employed differential gene expression analysis to understand the role played by the host genotype in determining thermal resilience as well as the impacts of different symbiont species. Investigating the dynamics of the host-symbiont relationship and response under heat stress is important for our understanding of coral resilience, but equally so for future attempts at assisted evolution.

## Results

**Heat stress-induced transcriptomic changes are host genotype-specific.** To investigate the overall effect of the different symbiont strains on the host's transcriptional response, we compared transcriptomic profiles under control and heat stress of all three host-symbiont combinations, used in this study, namely CC7, H2, and CC7-B01. In total, we detected the expression of 23,927 genes (93% of all gene models); of these 1179, 510, and 2039 genes were differentially expressed in CC7, H2, and CC7-B01, respectively, after 24 h of heat stress. A principal component analysis (PCA) was conducted to assess the differences in host-symbiont combination's transcriptomic profiles (Fig.1a). CC7-B01 clustered with CC7, revealing that transcriptomic profiles are highly driven by host genotype. Most importantly, although temperature-induced differences were evident, transcriptomes remained more similar between genotypes, even under stress. This suggests that the here-applied heat stress did not result in a strong common transcriptomic response, indicating that it may rather depend on the host genotypes capabilities (independent of associated symbiont).

To assess whether the associated symbiont type had an impact on the genotype's transcriptomic response, we compared the transcriptomic profiles of the two CC7 genotypes (Fig.1b). We observed clustering of the CC7 genotypes based on temperature conditions, regardless of their symbiont strain. CC7 and CC7-B01 appeared to cluster more closely under heat stress and control conditions. A Pearson's correlation analysis between CC7-B01 and CC7 at 32 °C ($p < 0.001$; $r^2 = 0.96$) and 25 °C ($p < 0.001$, $r^2 = 0.95$) showed a similarly high correlation maintained across conditions. The overlap of differentially expressed genes (DEGs) was relatively low between the three investigated host-symbiont combinations, with only 156 genes shared between them (79 up and 77 down-regulated; Fig. S1). Pairwise comparisons revealed that CC7 and CC7-B01 shared 2.5 times more DEGs than H2 and

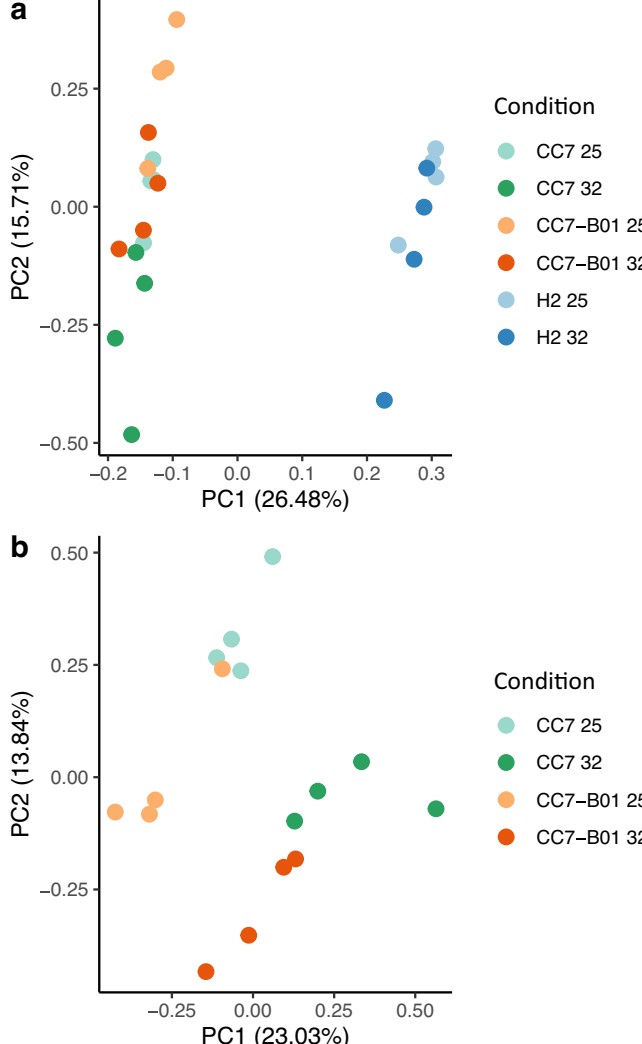

**Fig. 1 Principal component analysis of transcriptomic profiles at control and stress conditions. a** Analysis of all three samples transcriptomic profiles under control (25 °C) and heat stress (32 °C). **b** Analysis focusing only on CC7 and CC7-B01, to test the separation between control and stress.

CC7-B01. Interestingly, we found that the majority of DEGs detected in CC7-B01 were uniquely expressed in this host-symbiont combination. To learn more about differences and commonalities in transcriptomic change between the strains, we investigated the DEG response in further detail.

**Differences in thermal stress response of host genotypes.** The overall transcriptomic expression patterns were primarily host genotype-driven; however, DEG analyses indicated a potential influence of the symbiont. To investigate these, we performed a gene ontology enrichment analysis (GO-terms) on the DEGs derived from the three host-symbiont combinations (Supplementary Data S1, S2). We ran GO-term enrichment analysis on up-regulated and down-regulated genes separately for each strain. GO-terms showed a similar pattern to the results from the PCA (Fig. 1) in that CC7 and CC7-B01 shared more similarity in enriched terms (74 common terms) than they did with H2 (32 and 32 respectively). In all three strains, up-regulated genes showed enrichment of GO-terms relating to antioxidant response, metabolism, and unfolded protein response (UPR). Down-regulated gene enrichment showed no pattern in response, but

we noted some glucose-related GO-terms enriched in CC7-B01 (GO:0005355; glucose transmembrane transporter activity, GO:0005536; glucose binding, GO:0009749; response to glucose).

To better understand the nuances in the up-regulated GO-term enrichments of the strains, we focused on the GO-terms relating to biological function shared among all three host-symbiont combinations. Only 12 GO-terms were shared, which could be categorized into four groups (amino acid synthesis, mRNA splicing, oxidative stress, and protein folding response; Fig. 2). We observed that CC7 and CC7-B01 had strong enrichment in UPR, while oxidative stress response was most enriched in CC7-B01 and H2.

We investigated these observations on a gene-specific basis. We found that CC7 and CC7-B01 expressed temperature stress coping mechanisms such as heat shock protein 70 (HSP70), superoxide dismutase (SOD), thioredoxin reductase, and glutathione S-transferase (GST), which were significantly up-regulated in CC7 genotypes, but did not show any differential response in H2. Besides HSP70, CC7 also up-regulated protein chaperones and misfolded protein response genes such as 26S proteasome, calreticulin, and peptidyl-prolyl cis-trans-isomerase (PPI). Additionally, the CC7 genotype had increased expression of several genes with the same annotation, gene homologues that did not change in expression in H2. An important example of such is GST, which was not differentially expressed in H2 but had two up-regulated homologues in the CC7 genotypes.

The enriched oxidative stress response in the two strains sharing the same symbionts, CC7-B01 and H2, indicated that the symbiont might be exerting higher oxidative cellular stress on these hosts. Indeed, we detected that tumor-necrosis factor receptors and transcription factor AP-1, involved in early stress signals, showed significant up-regulation in CC7-B01 and H2, but not in CC7. Additional expression of specific heat stress genes previously reported in thermal response, such as nitric oxide synthase (NOS) and calnexin were significantly up-regulated only in CC7-B01. Interestingly, when we investigated the apoptotic response of each strain, we noticed that both CC7-B01 and CC7 appeared to be actively repressing apoptosis. Indeed, they both down-regulated caspase-3 and p53-inducible nuclear protein that controls autophagy, concomitantly they significantly up-regulated a putative BAX inhibitor. Additionally, we observed an impact on amino acid synthesis in CC7-B01 that was not evident in the homologous host-symbiont combinations.

Overall, the GO-term and gene-specific observations indicated the *B. minutum* symbionts shared by H2 and CC7-B01 may have been enhancing the overall stress experienced by the host. We then investigated to what extent the symbiont was impacting the host, and whether the transcriptomic predictions could be traced to a physiological response.

**Physiological responses of host-symbiont combinations.** Since oxidative stress can result in apoptosis, we investigated whether the increased level of oxidative stress had detectable adverse effects on cell survival in the cnidarian host. Through the measurement of caspase-3 activity, an apoptosis executioner, we tested host stress levels in the different host-symbiont combinations (Fig. 3).

Caspase-3 activation under temperature stress was significantly different between the three host-symbiont combinations, indicating differences in apoptosis response. Significant increases in caspase-3 in response to heat stress were only observed in CC7-B01 ($p < 0.01$) and H2 ($p < 0.001$). However, CC7-B01 also showed significantly higher caspase-3 activation than CC7 ($p < 0.01$) and H2 showed even higher activation than CC7-B01 ($p < 0.05$) during heat stress. These findings corroborated that the *B. minutum*

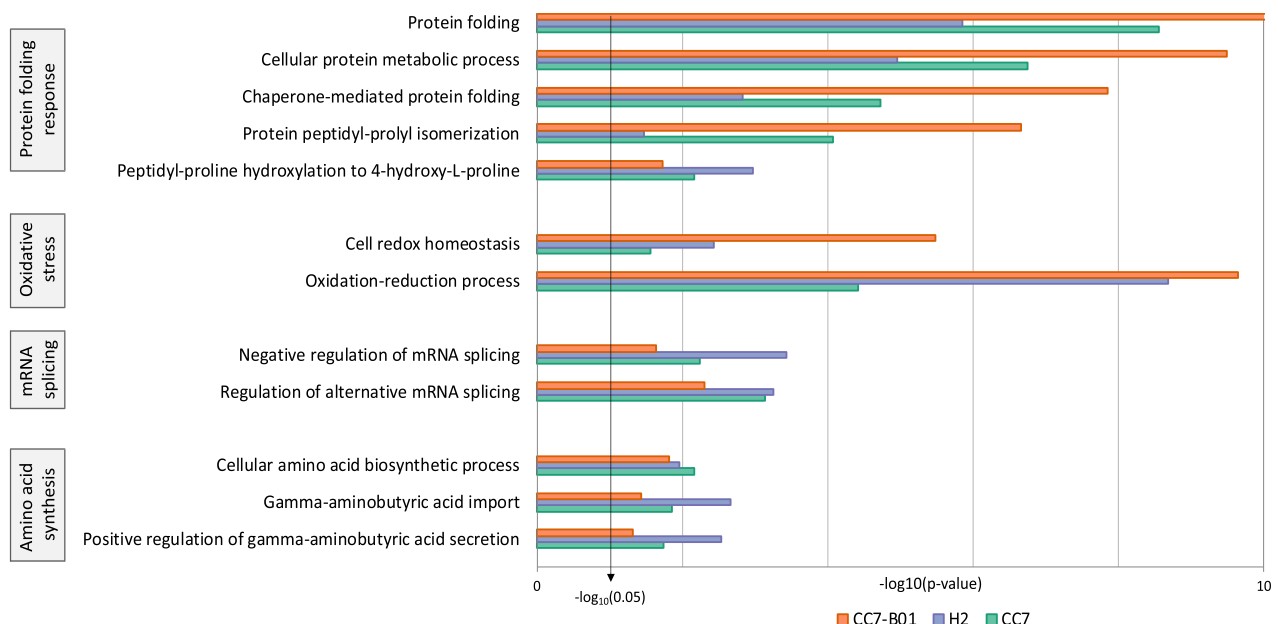

**Fig. 2 Common enriched GO terms of up-regulated transcripts in CC7, H2, and CC7-B01.** Analysis was focused on only biological function terms. These were separated into four pathways common in all three host-symbiont combinations as a response to heat stress. *P*-values of GO-terms are plotted on -log$_{10}$ scale to show enrichment strength ($n = 4$). All displayed terms are significantly enriched ($p < 0.05$); the more significantly enriched a term is, the larger the -log10 (*p*-value).

symbiont increases the stress experienced by the host, likely due to increased ROS production[26]. However, caspase activation in CC7-B01 at 32 °C was significantly different from CC7 ($p < 0.05$) and H2 ($p < 0.01$), placing it in between the two homologous host-symbiont combinations (Supplementary Data S3).

Since symbiont density plays a critical role in the stress experience of the host, and could influence observed caspase-3 activities, we counted *Symbiodiniaceae* cells per replicate and normalized the counts against total protein. Table 1 shows the normalized symbiont density, which revealed no significant differences between the two temperatures. However, we did observe significant differences between CC7 and CC7-B01 at 25 °C, with CC7-B01 exhibiting almost double the normalized symbionts counts ($p < 0.01$). Note that symbiont counts could not be conducted on the same individual before and after incubation time, due to the method requiring the anemone's homogenate. As such, the data shown here indicates that there was no significant difference between animals in the different treatment condition at the end of the incubation period.

**Heat stress affects host-symbiont metabolic compatibility.** The distinct apoptotic and caspase response of CC7-B01, as well as the gene expression changes in amino acid synthesis pathways observed, may be caused by the heterologous symbiont. We, therefore, determined if and how symbiosis was affected under heat stress in different combinations CC7, H2, and CC7-B01. Using a set of 731 previously identified Aiptasia symbiosis-associated genes[14], we investigated symbiosis maintenance in homologous and heterologous symbiont species (Supplementary Data 4). We found that 168 (CC7-B01), 94 (CC7), and 85 (H2) of these genes were significantly differentially expressed in response to heat stress in the different strains (Fig. 4). Additionally, we found that around 55% of symbiosis genes in CC7 and H2 were regulated in the opposite direction to what is expected in stable symbiosis. In contrast to CC7 and H2, CC7-B01 had significantly more genes (65%, $p < 10^{-6}$, chi-square test with Bonferroni

correction) showing changes in the opposite direction to what is expected in a stable symbiosis.

Part of the reversed symbiosis gene expression observed also involved critical processes in heat stress response, including HSP90, calumenin-B, and glutathione S-transferase. However, CC7-B01 stood apart from the homologous host-symbiont combinations by several genes relating to critical symbiosis nutrient exchange pathways: aldehyde dehydrogenase family members (AIP-GENE12723: fc −0.54; AIPGENE19640: fc −0.48; AIPGENE21619: fc −0.76), Niemann-Pick disease protein (NPC) (AIPGENE5532: fc −0.501), glutamine synthetase (AIPGENE26763: fc −0.28), gluta-mate dehydrogenase (AIPGENE26078: fc −0.33), and ammonium transporters (AIGPENE17420: fc −0.52; AIPGENE18105: fc −1.27) were down-regulated in CC7-B01. Interestingly, a number of these genes have been suggested to play a role in ammonium assimilation pathways in cnidarians. Thus, the downregulation of these in CC7-B01 suggests reduced ammonium assimilation in the host[14] (Fig. 5).

Genes involved in amino acid synthesis, such as betaine-homocysteine S-methyltransferase (BHMT: AIPGENE10977: fc 1.03), dimethylglycine dehydrogenase (DMGDH: AIPGENE10986: fc 1.4; AIPGENE10961: fc 1.35) and 4-hydroxyphenylpyruvate dioxygenase (HPD: AIPGENE25576: fc 0.65) also displayed expression changes in the opposite direction compared to stable symbiosis in CC7-B01. Indeed, during stable symbiosis these genes are expected to be down-regulated and/or with no significant change in expression as was the case for H2 and CC7. Overall, CC7-B01 showed a general up-regulation of the sulfur-containing amino acid synthesis pathway, which during stable symbiosis, is otherwise down-regulated.

**Discussion**

In this study, we examined the transcriptome-wide responses of a symbiotic cnidarian to acute heat stress and investigated response differences between homologous and heterologous host-symbiont combinations. Due to Aiptasia's symbiotic flexibility, we were able to assess not only cnidarian host-genotype driven responses, but

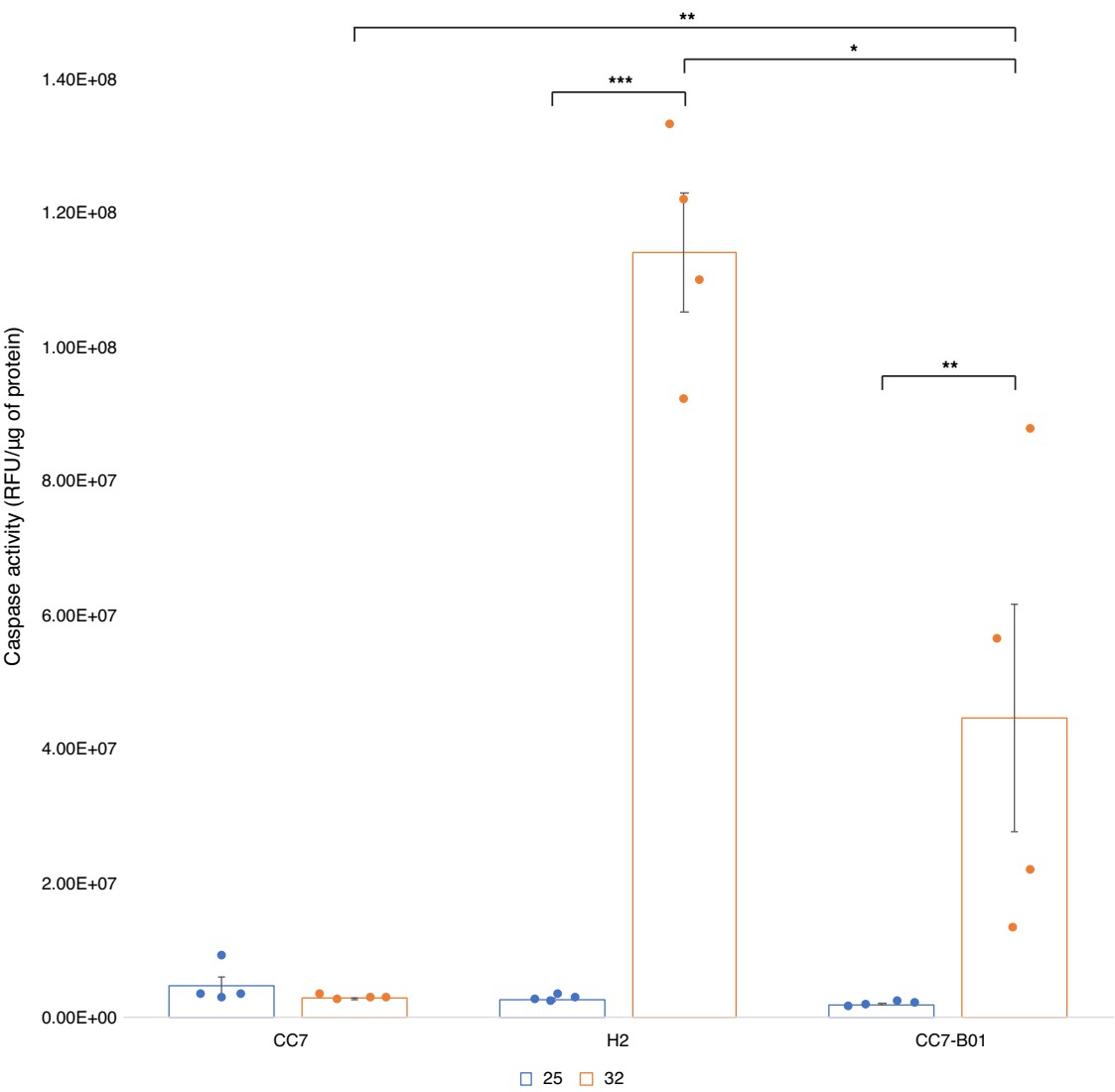

**Fig. 3 Caspase-3 activation at control and after heat stress in CC7, H2, and CC7-B01.** Blue and orange bars are caspase-3 activation at control and heat stress exposure after 24 h, respectively, with standard error ($n = 4$). Significant differences are denoted with asterisk (* $= p < 0.05$ ** $= p < 0.01$, *** $= p < 0.001$) and absence indicates no significant differences. Caspase activity was measured in relative fluorescent units and normalized per unit of protein (mg).

**Table 1 Normalized symbiont density ($n = 4$ per strain-temperature combination).**

| Strains | 25 Degrees | | | | 32 Degrees | | | | p value |
|---|---|---|---|---|---|---|---|---|---|
| | rep1 | rep2 | rep3 | rep4 | rep1 | rep2 | rep3 | rep4 | |
| CC7 | 13,435 | 10,703 | 10,314 | 10,994 | 17,107 | 20,738 | 10,152 | 28,091 | 0.091 |
| H2 | 17,327 | 11,896 | 9987 | 9090 | 14,777 | 14,567 | 1197 | 13,656 | 0.424 |
| CC7-B01 | 27,811 | 17,044 | 28,412 | 18,159 | 11,808 | 26,015 | 10,951 | 24,080 | 0.34 |

Units are counts [ml filtrate]$^{-1}$[g Aiptasia protein]$^{-1}$. No significant changes in symbiont counts were detected between treatments (two-tailed t-test).

also to evaluate the influence of the symbiont. Furthermore, we were able to assess the regulation of symbiosis mechanisms between different host-symbiont combinations under heat stress. We show that not only are the dynamics of the cnidarian-algae symbiosis complex, but also that cnidarians harbouring heterologous symbionts respond differently to heat stress than those associated with their homologous Symbiodiniaceae strain. This finding was particularly exacerbated for genes that are essential to symbiosis.

The cnidarian host plays an important role in determining the success of the holobiont under stress[44,45]. While a core cnidarian heat stress response exists, variations are evident between species and even genotypes[26,46,47]. Notably, the results indicate that splicing-related genes are differentially regulated across the strains investigated. Splicing repression can act as a form of protein production control in eukaryotic cells that allows for the selective expression of proteins in response to stress[48], and the expression of related genes has previously been reported in coral

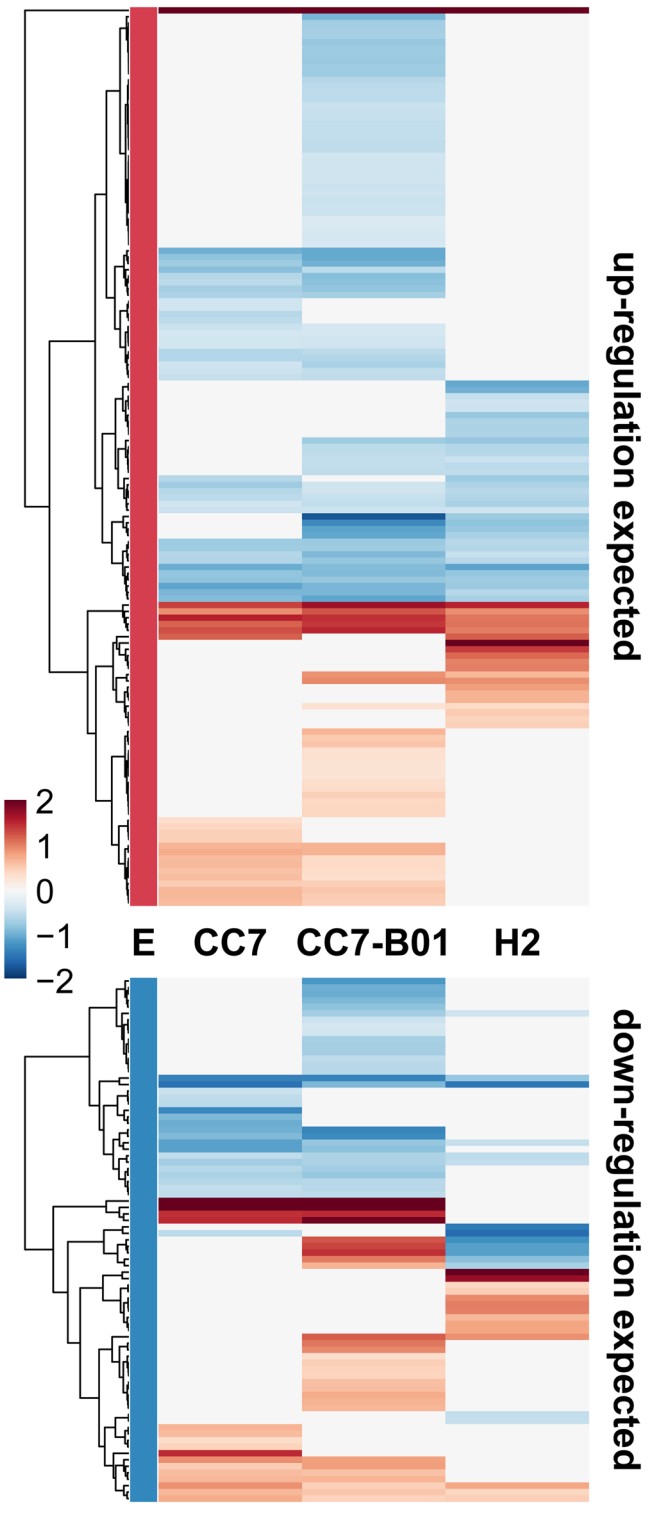

**Fig. 4 Heat map indicating expression changes of symbiosis-associated genes in response to heat stress.** The three strains are labelled accordingly. The column labelled as 'E' represents the expected gene expression changes in known symbiosis-associated genes from Cui et al., 2019. Genes in E are coloured by expected responses to symbiosis: up-regulation (top half of the figure; red E) or down-regulation (bottom half; blue E) during symbiosis relative to aposymbiotic state. Other columns indicate observed responses in each of the three studied strains. Coloured columns indicate significant differential expression ($n = 4$; $p < 0.05$; red is up-regulated, blue down-regulated); clear columns indicate no significant change. Values of colour scale are $\log_2$-transformed fold changes (fc).

heat stress studies[26,49]. Here, we clearly show that host genotypes have distinct transcriptomic profiles. Elevated temperatures did not lead to a higher similarity between transcriptomic profiles of CC7 and H2. Most importantly, CC7 and CC7-B01 maintained a strong similarity, despite carrying different symbiont species. Distinct differences between CC7 and H2 genotypes were maintained throughout control and temperature stress, suggesting that functional requirements and response mechanisms are different.

While we recognize that endogenously H2 and CC7 are associated to different types of symbionts, the up-regulation of pathways crucial to an efficient stress response in CC7 suggests that this genotype has a superior thermal response compared to H2, which is in line with a previous study comparing the physiological responses of these two strains to increased temperatures[36]. One of these pathways is the antioxidant response, evident through the activation of coping mechanisms such as SOD and GST[50]. Recent studies have shown that heat shock can induce antioxidant mechanisms and production of other proteins aiding in stabilizing nascent proteins, which is also indicative of endoplasmic reticulum stress[51]. Other upregulated pathways in CC7 genotypes include protein chaperones and misfolding response mechanisms, such as HSP70 that has been recognized as a distinguishing factor between corals that are more or less thermally tolerant[40,52]. Both protein folding and antioxidant mechanisms have repeatedly been reported as part of an efficient heat stress response[47,53,54]. Additionally, apoptosis regulation and inhibition in CC7 and CC7-B01 indicated an active pro-survival response to heat stress[55,56]. Indeed, testing of the observed apoptosis-related transcriptional response showed that caspase-3, which is considered the most important executioner of apoptosis[57], corroborated our hypothesis. H2 exhibited significantly higher caspase-3 activation compared to CC7. Although CC7-B01 displayed significantly less caspase-3 activation than H2, it was significantly higher than CC7. This indicates that thermal resilience of the associated symbiont can impact the level of stress and response of the host.

Symbiont species can differ in temperature resilience. *B. minutum*, for example, are typically less thermotolerant than other Symbiodiniaceae strains[58]. H2-associated symbionts have a higher production of reactive oxygen species (ROS) in response to heat stress than those associated with CC7[26]. The excess ROS can potentially leak to the cnidarian host cells, resulting in oxidative stress[25,59,60]. A high abundance of ROS and consequentially oxidative stress can cause cell damage and potentially trigger the initiation of apoptotic pathways[61,62]. Hence, Symbiodiniaceae thermal resilience can affect the response and overall resilience of the host coral[23,63–65].

Our results show that the symbiont's thermal resilience is a critical factor in the physiological stress experienced by the host, but that the host itself must also exhibit thermal resilience in order to provide a suitable environment for the symbiont. In the case of CC7 and CC7-B01, this leads to the subtle, yet important differences observed. We found that oxidative stress response, as well as early stress response pathways identified in corals[54], were significantly affected in strains H2 and CC7-B01 harbouring *B. minutum*. The impact of the symbiont on the host was further evident in the significant differences in caspase-3 activity. CC7-B01's activation levels were significantly different from both H2 and CC7. This highlights the importance of the host genotype and its ability to mitigate additional stress imposed by the symbiont. Our results corroborate the findings presented by Herrera et al. (2021) and confirms that the host's thermotolerance is a determining factor in ensuring holobiont resilience by providing a suitable environment in which the symbiont can perform optimally.

Our results show that the CC7-B01 changed the expression of genes involved in important nutrient pathways, opposite to what is observed under stable symbiosis. As observed in other studies,

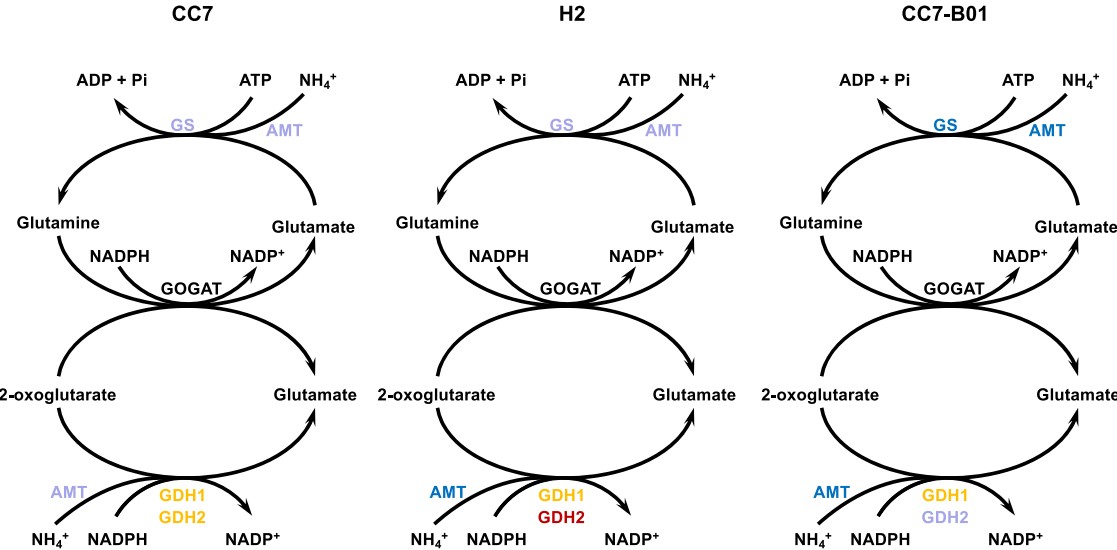

**Fig. 5 Ammonium assimilation pathway expression changes after heat stress in CC7, H2, and CC7-B01.** The most significant response changes were observed in CC7-B01, where most genes were significantly down-regulated (when they were expected to be up-regulated in a stable symbiosis). GS glutamine synthetase, AMT ammonium transporter, GOGAT glutamate synthase (was not detected in transcriptomes), GDH glutamate dehydrogenase. Colour code shows light purple–down-regulated, dark blue–significantly down-regulated, yellow–up-regulated, red–significantly up-regulated.

non-native host-symbiont pairings may result in sub-optimal nutrient exchange under normal conditions[66], which is likely to worsen under thermal stress. A recent study on CC7, H2, and CC7-B01 showed that the rate of carbon and nitrogen assimilation was significantly different between homologous and heterologous host-symbiont. Following incubations with stable isotope tracers, Symbiodiniaceae types exhibited stable enrichment levels regardless of host identity. While the enrichment in the tissues was host-dependent, the heterologous association contained less carbon and nitrogen in its tissue, most of the carbon enrichment was instead found in the symbiont[67]. Given that symbiont densities were higher in the heterologous host symbiont combination CC7-B01, the differences in host carbon enrichment were likely due to differences in photosynthate translocation rates. This suggests that reduced metabolic stability and consequent incompatibility already occur under optimal conditions and are further exacerbated under increased temperatures. It is important to restate here that we also observe the same increased symbiont density in CC7-B01 at 25 °C as reported by Rädecker et al.[67]. This increased symbiont density is likely a consequence of the reduced translocation of photosynthates from the symbiont to the host, which lowers the host's ability to assimilate waste ammonium and thus increases the availability of nitrogen to the symbionts[14].

The nutritional exchange in cnidarian-algae symbiosis is based on the translocation of photosynthates from the symbiont to the host. In return, the host provides nutrients, such as nitrogen, to the symbiont. Photosynthesis-derived carbon, in the form of glucose provided by the symbiont, has been suggested to regulate pathways of ammonium assimilation, allowing the host to control the nitrogen flux to the symbiont[14,68,69]. However, we identified downregulation of symbiosis-specific ammonium transporters and assimilators during heat stress (Fig. 5) in CC7-B01. Reduced assimilation of ammonium in the host cell tissue allows an increased flow of nitrogen to the symbiont, therefore disrupting the nitrogen-limited state characteristic of stable symbiosis[13,15]. If symbiont-derived carbon drives host ammonium assimilation, then the observed reduction in genes involved in ammonium assimilation also reflects reduced photosynthate translocation

and, therefore, a metabolic decoupling between partners. Indeed, CC7-B01s showed significant down-regulation of aldehyde dehydrogenase, an enzyme suggested to play a role in carbon incorporation into lipid bodies[70–72], and of cholesterol transporters[9,73], both reflecting limited glucose-derived carbon availability to the cnidarian host. Furthermore, CC7-B01 showed a systematic up-regulation of genes involved in choline-betaine pathways that are otherwise typical for aposymbiotic Aiptasia, depending on heterotrophic feeding[14]. Upregulation of BHMT, DMDGH, and 4HPD enzymes involved in amino acid synthesis from food-derived nutrients indicate that CC7-B01 holobiont have an increased demand for heterotrophically acquired choline to satisfy its nutritional needs (Fig. 6). The increased dependence on exogenous sources for amino acid synthesis further highlights that availability of glucose-derived carbon from the symbiont is limited, reducing the host capacity to assimilate ammonium, and reverting to synthesis pathways characteristic of aposymbiotic animals. To compensate for such reduced photosynthetic carbon availability, the cnidarian host will likely have to rely on the utilization of storage lipids and heterotrophy to fulfil its metabolic energy demand during stress[15].

Overall, heat stress-induced changes in gene expression in CC7-B01 indicate that this host-symbiont combination reaches metabolic instability and, hence, bio-energetic disruption of symbiosis, earlier than homologous combinations CC7 and H2. This is reminiscent of findings in studies using heterologous symbionts in Aiptasia and the corallimorpharian *Ricordea yuma*, which observed transcriptomic responses intermediate between symbiotic and aposymbiotic individuals and suboptimal nutrient exchange[74,75]. The fact that CC7-B01 carries the same symbiont as H2, which is the homologous host of the B01 symbiont, therefore, further suggests that successful and resilient symbiosis relies on the physiological compatibility of host and symbiont and echoes previous findings in Aiptasia[36,76]. This compatibility is likely a product of co-evolutionary processes between both partners as well as any adaptations to their local environment.

However, the role of the host and algal symbiont in determining the holobiont's thermal resilience is challenging to fully

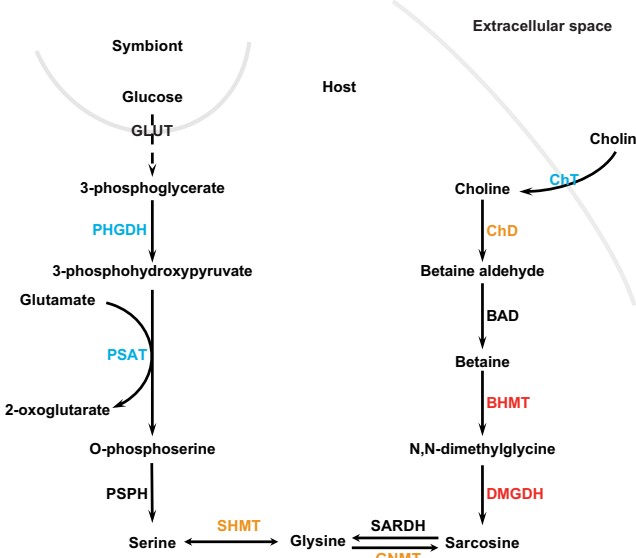

**Fig. 6 Pathway for glycine and serine biosynthesis in CC7-B01 response to heat stress.** In a stable symbiosis, symbiont-produced carbohydrates are a main source of carbon backbones for the synthesis of nonessential amino acids in the cnidarian host (left side). In contrast, pathways using food-derived choline and betaine for amino acid biosynthesis are down-regulated (right side). During stress exposure, the stable state of the symbiosis shifts to a carbon-limited state, resulting in increased dependency of the host on external carbon sources. In response, the amino acid biosynthesis pathway using food-derived carbon sources is now up-regulated, while the pathway relying on symbiont-derived glucose is down-regulated. Colours represent the direction of expression change: dark red–significantly up-regulated, orange–up-regulated, dark blue–significantly down-regulated, light blue down-regulated (GLUT–glucose transporter, PHGDH–phosphoglycerate dehydrogenase, PSAT–phosphohydroxythreonine amino- transferase, PSPH–phosphoserine phosphatase, SHMT–serine hydroxymethyltransferase, ChT–choline transporter, ChD–choline dehydrogenase, BAD–Bcl2 associated death promoter, BHMT–betaine—homocysteine S-methyltransferase, DMGHD–dimethylglycine dehydrogenase, GNMT–glycine N-methyltransferase, SARDH–sarcosine dehydrogenase)[14].

untangle. Our results provide evidence of both host- and symbiont-specific adaptations and, importantly, highlight the significance of the interaction between the two. Keeping in mind the distinct origins of the CC7 and H2 genotypes, CC7 may have acquired thermal resilience due to its natural habitat in Florida, where strong seasonal fluxes and higher maximum temperatures occur[26,77]. This suggests that the ability of host-symbiont pairings to their local thermal profile may promote better coping mechanisms. While the host's response seems to be crucial in dealing with imposed stress, we show that the symbiont's thermal resilience plays a role in the overall physiological stress experienced by the host.

The stress experienced by coral host and symbiont is known to affect the sensitive metabolic equilibrium that underpins the symbiosis. Our results show that under stress, the heterologous host-symbiont combination displays significant destabilization of essential nutrient cycling pathways in the symbiosis earlier than homologous counterparts. Shifts in nutrient exchange and assimilation indicate that the heterologous host-algae relationship is more susceptible to heat stress than homologous ones, likely due to co-evolutionary processes in homologous host symbiont pairings. In extension, heat-resilient symbionts may not be easily interchangeable, even within different populations of the same

cnidarian species, as they might not match the host's physiology. Our study shows that the selective and evolutionary conserved cnidarian-algae relationship might strongly rely on the physiological compatibility of both partners. Achieving and maintaining metabolic stability is important for the long-term success of a stable symbiotic relation.

## Methods

**Growth and maintenance of Aiptasia strains.** Two Aiptasia strains were used in this study: CC7 originated from Florida, while H2 was originally collected from Coconut Island in Hawaii. Both were courtesy of the Pringle Lab (Stanford University, CA, USA). All Aiptasia strains were housed in separate polycarbonate tanks and incubated at 25 °C at a light intensity of 80 μmol photons $m^{-2} s^{-1}$ on a 12 h:12 h light: dark schedule. The Aiptasia were fed with freshly hatched *Artemia salina* (brine shrimp) nauplii thrice per week.

**Bleaching and re-infecting Aiptasia.** In order to combine CC7 host with H2-derived symbionts, we first bleached CC7 by menthol treatment, as described in Matthews et al.[42]. The treatment was repeated until complete bleaching was observed. The aposymbiotic state of CC7 was further confirmed visually with a fluorescence-stereomicroscope. Bleached anemones were then kept at 25 °C in the dark for >12 months, with a feeding schedule maintained the same as for symbiotic anemones.

Subsequently, the aposymbiotic anemones were infected with Symbiodiniaceae strain SSB01, which was previously isolated from H2[78], and after that maintained under regular culture conditions as the other Aiptasia strains (>12 months). The successful reinfection of the CC7 anemones with SSB01 symbionts was validated through PCR amplification of ITS2 regions[24]. The resulting combination (CC7 host with SSB01 symbionts) was called hereafter CC7-B01.

**Subjecting Aiptasia to heat stress.** Twelve anemones of each strain were taken from two separate stock tanks and relocated into new polycarbonate tanks to account for batch effect. Six individuals were used per treatment to ensure enough replicates if accidental death should occur under stress conditions. The tanks were incubated without food for three days to allow the Aiptasia to settle and acclimatize to their new tanks. Except for the temperature, all conditions (e.g., size of the tank, degree of illumination, and light: dark cycle) were identical between heat-stress and control treatments. Control tanks remained in the 25 °C incubator. Heat stress tanks were placed in an incubator that slowly ramped up from 25 to 32 °C at the rate of 2 °C per hour starting from 8 am and reaching the target temperature by noon. The stress duration (24 h) lasted from noon until noon the next day.

**Extraction of RNA from Aiptasia anemones.** Four individual biological replicates were picked from control and treatment tanks. Individuals were placed in separate Eppendorf tubes, dried by removing excess water and weighed. Aiptasia were of comparable sizes across strains. Buffer RLT was added proportionally to the weight of each sample, and the Aiptasia was crushed using a motorized pestle with Kontes RNase-free tips (Kimble Chase, Vineland, NJ). RNA was extracted using the RNeasy Mini Kit (Qiagen, Hilden, Germany). The RNA extraction procedure was conducted as per manufacturers protocol and ultimately eluted in 50 μl of RNase-free water. RNA concentration of the samples was quantified using Qubit 2.0 (Invitrogen, Carlsbad, CA), and quality-checked using Bioanalyzer 2100 (Agilent, Santa Clara, CA).

**Library construction from total RNA and transcriptome analysis.** Total RNA from all samples were initially subjected to a poly-dT selection step and then used to generate libraries using the Illumina TruSeq RNA Sample Prep Kit (Illumina, San Diego, CA) per manufacturer's instructions. Sequencing was carried out on the HiSeq 2000 platform (Illumina, San Diego, CA), producing a total of 874 million reads.

Reads were mapped to gene models from Aiptasia[42] with kallisto v0.42.4[79]. To reduce ambiguity in read-mapping downstream, duplicate gene models were removed as mentioned previously in[26]. Using TPM values produced from kallisto, we identified genes that were differentially expressed (Wald test, adjusted $p$-value <0.05) under heat stress relative to control conditions by using sleuth v0.28.0[80].

Functional enrichment was carried out using topGO v2.28.0 (Alexa et al., 2006) with default settings. GO terms with $p < 0.05$ and occurring ≥5 times in the background set were considered significant.

**Physiological measurements.** A follow-up experiment with the same procedures as before was conducted to test for caspase-3 activation: twelve anemones for each strain taken from two separate tanks, with four individuals used per treatment. After 24 h of heat stress, anemones were placed in individual 1.5 ml Eppendorf tubes and washed twice in PBS. Anemones were then crushed in 400 μl of cell lysis buffer (200 mM TRIS- HCl pH 7.5, 2 M NaCl, 0.1% Triton 20%). Homogenates were spun down at 14,000 $g$ for 3 min, and 100 μl of supernatant were extracted for caspase-activation measurements. Activation was measured using EnzChek Caspase-3 Assay Kit 2 (Thermo Fisher Scientific, Massachusetts, USA) following

the manufacturer's protocol. The assay measures caspase-3 activity, which includes caspase-3 and other DEVD-specific activity. To account for differences in cell numbers due to anemone size, total protein content was measured using 200 μl of supernatant using Micro BCA Protein Assay Kit (Thermo Fisher Scientific) according to the manufacturer's protocol. Caspase value was normalized against total protein content. Change in caspase activation was assessed using t-tests with Bonferroni correction. Symbiodiniaceae cells were counted with Guava flow cytometer. For this, a total of 25 μl of the anemone homogenate was sheared through a 25-gauge needle affixed to a 1 mL syringe and diluted in 225 μl of SDS 0.1%.

**Reporting summary**. Further information on research design is available in the Nature Research Reporting Summary linked to this article.

## Data availability

RNA-seq data are available at NCBI under project number PRJNA406873. All other data are available from the corresponding author (or other sources, as applicable) on reasonable request.

## Code availability

Code, read count data and intermediate files are available at https://github.com/lyijin/increased_incompatibility.

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

## Acknowledgements

We thank Craig Michell for RNAseq library preparation, Alicia Schmidt-Roach for anemone husbandry and Jean-Baptiste Raina for feedback on the manuscript. Research reported in this publication was supported by baseline funding from KAUST to M.A.

## Author contributions

M.A. conceived the idea and obtained funding. M.J.C., Y.J.L., and M.A. designed experiments. M.J.C. and Y.J.L. conducted heat stress laboratory experiments. M.J.C., Y.J.L. and G.C. conducted transcriptome analyses. M.J.C. wrote first draft of the manuscript. All authors approved the final manuscript.

## Competing interests

The authors declare no competing interests.
