## [Peer Review File · Communications Biology]

Reviewers' comments:

Reviewer #1 (Remarks to the Author):

This manuscript broadly addresses the effects of thermal stress on transcription and caspase activation in the cnidarian-dinoflagellate symbiosis using the model symbiotic sea anemone *Aiptasia*. Specifically, it examines the interaction between host and symbiont genotype under heat stress. Importantly, it compares the profiles of sea anemones and their homologous symbiont species to a commonly used heterologous combination of CC7 and the algal strain SSB01 (*Breviolum minutum*). This pairing of CC7 and this heterologous symbiont has been used in many studies as a tool to model a healthy symbiosis state, indeed the standard, for the transcriptomic study of symbiotic conditions in *Aiptasia*. However, in this study, the authors find that this heterologous association displays transcriptomic hallmarks of metabolic incompatibility. Together with transcriptomic data, the authors also further examine apoptotic response to heat stress using a caspase assay and show activation in heat-susceptible combination of H2-B. *minutum* and CC7-B. *minutum*, while no response is found in CC7-S. *linucheae*. Finally, the differential expression from the heat stress dataset is viewed in the context of a set of upregulated/downregulated symbiotic state genes generated by a previous meta-analysis.

This work is composed of a straightforward analysis of RNA-Seq data, coupled with a follow-up caspase-like activity assay and a comparison to a previous study that generated a symbiotic state gene list. Overall, the paper is well-written and clearly understood. Methods are clear but brief and figures communicate findings well. Comparing the transcriptomes of homologous combinations to heterologous combinations of *Aiptasia* under heat stress is novel and of broad importance to the coral symbiosis field, with implications for past and future studies that use *Aiptasia* and specifically CC7-SSB01 partners as a symbiosis model. The work accompanying the transcriptomic dataset fits into the story, but the figure presentation occasionally feels incomplete. More transparency and detail is needed in general to allow for reproducibility.

In conclusion, this manuscript will help the coral symbiosis field understand how host genotype and symbiont identity influence response to heat stress, two areas of research that are critical to understand in the face of coral decline due to climate change. The conclusions drawn from this paper mesh well with previous studies of heterologous associations in *Aiptasia* and extend evidence of symbiotic incompatibility into thermal stress conditions.

I have several general points as well as some specific notes on lines in the text.

Comment 1: The use of homologous and heterologous should be defined and justified early in this paper. Often heterologous is used to describe partners that are non-native to their host species. In *Aiptasia* this has often been with species such as *D. trenchii* or *S. microadriaticum* which are never found in the host in nature. However, here both species *B. minutum* and *S. linucheae* are found in naturally occurring populations of *Aiptasia*, they just differ based on geographical location.

Comment 2: With respect to the methods, I found myself needing clarification in several places, which I will separate out for ease of addressing:

2a: Paragraph at Line 123: In this study the authors compare sea anemones with their homologous symbiont to sea anemones that were bleached and recolonized with SSB01. Was there evidence that all three combinations had roughly the same symbiont density and were at the same stage of symbiosis (e.g. not still in the process of recolonization)? Is there any visual or cell counting information to show that algal densities were roughly similar between all three combinations, since one originated from aposymbiotic animals? This would be useful information to have, as other studies of *Aiptasia* had heterologous colonization for one year and found much lower symbiont density in the heterologous pairing (Medrano et al., 2019). If no data is available, there at least is some reference to equivalent densities buried in the discussion at line 410. At the very least I would expand on this statement in this method paragraph.

2b. Paragraph at line 151: please include accompanying scripts in the supplemental data or at another specified repository for best practices of data reproducibility.

2c. Paragraph at line 165: Please include the sample size for this experiment. If sample number is the same as the previous experiment be more explicit. Also, this paragraph indicates Symbiodiniaceae were counted at line 177, but I can't find the corresponding symbiont count data in the results section. This would be useful to see any differences in symbiont density. In addition, the volumes of lysed homogenate used for the caspase assay and for the protein assay should be included. It should further be noted in the methods that this assay measures caspase 3-like activity, including caspase-3 and other DEVD-specific activity.

Comment 3: I have a general question regarding whether differences seen between CC7 and H2 are biological as stated or if there are differences related to technical processing:

3a: For example, it would be useful to have a supplementary table showing specific information for CC7/H2/CC7-B01 on read counts, percentages of successfully mapped reads, and resulting gene model coverages. Or simply include text in the results e.g. at line 185 that briefly states read counts, mapping and gene model coverage were similar between pairings. The reason why I am concerned is the presentation of Figure 4/Table S3. Figure 4 has me concerned that transcriptome profiles for H2 samples are narrower/shallower than CC7 samples. If so this could mean that the lower amount of DE genes from H2 could be partially attributed to technical difference rather than biological difference. Since reads are mapped to Aiptasia gene models created from CC7 genotype, evidence or lack of evidence for this mapping bias would be very useful to know with regards to Figure 1, 2, and 4.

3b. Figure 1 is clearly presented and described. The authors explain how genotypes clearly differ, and how CC7 genotype changes significantly with respect temperature stress. H2 genotype does not cluster separately but it is unclear whether that difference comes with the caveat of restricted overall transcriptomic profiles.

Comment 4: Figure 2 is also clearly presented. Since there is no mention of mRNA splicing yet it is 25% of the figure, I think readers would benefit from the inclusion of a short blurb about its continued presence in coral symbiosis heat stress studies similar to the paragraph found in Cziesielski et al. 2018 <http://dx.doi.org/10.1098/rspb.2017.2654>.

Comment 5: Figure 3 should have proper units on the y-axis, with μmol of product (R-110) per unit time (h) per unit of protein (mg). If no reference standard was used, then signify arbitrary units in the legend. Currently it appears as if fluorescence is normalized to total protein per animal, but it would make more sense to normalize by the amount of protein present in the equivalent volume used for the caspase assay. Was the inhibitor from the kit used as a control to verify that the fluorescence signal is specific to caspase? If so, I would include that data.

Comment 6: Figure 4 is very interesting but difficult to read. It would benefit from having clear titles for the separate upregulated and downregulated sections. The column labels are fine in the middle, but the figure would benefit from labels being placed on the top as well. The legend should make clear the organizational premise of the figure, that column E is a gene list from a previous study identified as upregulated or downregulated during symbiosis. The legend should also indicate how significant fold-change was determined.

Comment 7: I have a bit of confusion with the underlying data of Figure 4 and Table S3.

Comment 7a: As its written in Figure 4, "clear columns represent no significant fold change between treatments". In the accompanying supplemental data Table S3, a majority of the genes show a fold change of exactly "0". Does this mean they were below the significance cutoff? If so this should be stated in Table S3. However, at the top of Table S3 the qualification is "at least one strain has to have the gene expressed to be considered". Does this mean the gene was not present in that sample's transcriptome? If so, it needs to be clarified in Figure 4 and Table S3 that white areas or "0"s indicate absent data, and not true measurements of fold change.

Comment 7b: It would be useful for Table S3 to include gene annotations, so readers don't have to switch over to Table S1 to gain information. As well as the significance cutoff or cutoffs used to

categorize genes as significantly upregulated or downregulated.

Comment 8: (Line 320) Regarding the Figure 5 legend: the wording of the parenthetical is ambiguous. Was the expectation to be upregulated in heat stress or in symbiotic state? Overall, I think the figure and the rest of the text does a good job showing that ammonium assimilation is reduced more under thermal stress in heterologous combination.

Comment 9: Regarding Figure 6, the callback to the Cui et al. 2019 figure is well done and helps show exactly how heat stress alters these two pathways during symbiosis.

Comment 10: Regarding CC7 and its geographic location. CC7 originates from an Aiptasia population sent to the Pringle lab from a North Carolina company (Carolina Biological) (Sunagawa et al. 2009). This is different than being collected on the North Carolina coast, which is much more temperate and where Aiptasia contains *B. psysgmophilum*, a cold-tolerant symbiont found along the eastern coast of the United States. *S. linucheae* is thermally tolerant to heat but has only been collected from the Caribbean previously (Diaz-Almeyda et al 2017, https://www.algaebase.org/search/species/detail/?tc=accept&species_id=145014). To my knowledge *S. linucheae* samples have not been collected off of North Carolina. However, it is commonly found as a symbiont to *Exaiaptasia* found in Florida (Medrano et al. 2019). This corresponds with CB website <https://www.carolina.com/marine-and-saltwater-animals/sea-anemone-aiptasia-living/162865.pr> which says they ship anemones directly from a vendor on the Florida coast. All this to say, because of CC7 unknown origins, it is difficult to say how thermally distinct the original locations of the anemone clonal lines are. This is important for the rest of the paper especially when attempting to compare relative thermal regimes of locations. CC7 is a fantastic tool to study symbiosis but the strain should not have ecological meaning thrust upon it. This location name propagates through studies and misinforms readers. Below I've included lines where reference to this geographical location may need to be altered or clarified.

Line 16: "two thermally distinct locations, North Carolina (CC7) and Hawaii (H2)."

Line 95: "distinct locations, North Carolina (CC7) and Hawaii (H2),"

Line 112: "CC7 originated from North Carolina"

Line 473: "due to its natural habitat in North Carolina"

Comment 11: The discussion of this work and its comparison with some previous homologous v heterologous studies in Aiptasia and other species is lacking. For example, Matthews et al. 2017 (10.1073/pnas.1710733114) shows a similar heterologous transcriptomic shift in "stable symbiosis", and Medrano et al. 2019, which uses CC7 and *D. trenchii* as partners, shows similar patterns in its protein dataset (10.3389/fmicb.2019.01153). Lin et al. 2019 also shows transcriptional differences between homologous/heterologous symbionts with a corallimorpharian (doi.org/10.1242/bio.038281).

Comment 12: Bibliography has several errors that require correcting and that result in numerous in-text citation problems (e.g., at lines 32, 55, 80, 379, 384).

Minor comments:

Line 92: The species name of Aiptasia is never used in the manuscript and should at least occur here in parentheses as "Aiptasia (*Exaiaptasia diaphana*)".

Line 96: fix spelling of "Symbiodinium *linucheae*"

Line 143: Can the authors provide an estimate of animal size (e.g. oral diameter)? There is currently no reference here for what "larger" entails.

Line 158: fix spelling of "duplicate"

Lines 160: Please add version information to sleuth. As well as significance cutoffs for differential expression analyses.

Line 162: Please add version information for topGO and specify what default generally entails.

Line 171: fix spelling of "EnzChek"

Line 375: Symbiont species can differ in temperature resilience.

Line 387: change "himself" to "itself"

Line 397: I don't know what "exasperated" means in this context.

Reviewer #2 (Remarks to the Author):

This study by Cziesielski and colleagues investigates the role of both host and algal symbiont in thermal tolerance of the cnidarian holobiont. The authors use the fact that *Aiptasia* is compatible with multiple strains of symbiotic algae and constructed a heterologous combination between CC7 anemones and SSB01 (derived from H2 anemones) and then used RNA-sequencing to compare the transcriptional profiles of each strain under control and thermal-stress conditions. The study found that CC7 and CC7-SSB01 has similar transcriptional profiles and that CC7 and H2 remained distinct throughout the thermal stress experiment. While symbiosis-associated genes were differentially expressed in all strains in response to heat, the CC7-SSB01 combination showed a more dramatic response, potentially suggesting that the heterologous nature of that symbiosis was less stable than those of CC7 and H2 with their endogenous symbionts.

This study is both important and timely in order to better understand the dynamics of symbiosis overall in the cnidarian-dinoflagellate system and especially under thermal-stress. The manuscript is well written and easy to follow and read. The conclusions reflect the data generated from the experimental design (with the noted exception below). The statistics and bioinformatics are justified and explained clearly with appropriate citations.

The work provided here is convincing based on the data presented. However, the one addition that could have made this comparison across strains stronger would be to have the reciprocal heterologous symbiosis with H2 anemone with SSA01 algae derived from CC7 anemones. It is a severe limitation of this study to only have CC7 with the heterologous algal strain. The authors make conclusions that CC7 may be the more thermally-tolerant anemone due to the results of CC7-SSB01 and compared to H2 but of course, the difference between CC7 and H2 is not just the anemone but also the algal species harbored endogenously by both anemones. This study still provides useful insights into the dynamics of symbiosis and especially under thermal stress. To be more complete, either more strains of anemones or more heterologous combinations could have been included. This is not a fatal flaw but this type of comparison needs to be done to really tease out the role of both host and algal symbiont in thermal tolerance of the holobiont. The RNASeq addition here provides a nice snapshot at gene expression during thermal stress, however a time series approach would have also identified genes that are differentially expressed in the strain combinations before, during, and after bleaching. As it is, there is no clear mention in the text when the anemones were sampled for RNA extraction after being exposed to heat stress. "The heat stress duration (24 hours) lasted from noon until noon the next day". So, anemones were sampled immediately after? We know from Cleves et al. 2020 (PNAS) that there are strongly differentially expressed genes very soon into the heat stress and return to baseline well before bleaching actually occurs. So, picking one time point and making lots of conclusions based on genes that are differentially expressed at that point is difficult.

More detailed/specific comments:

Throughout the draft, there are numerous inconsistencies with the in-text citations. In many cases, the author list needs to be truncated to "et al." after the first author. I think this is a result of the citation management software, but this needs to be corrected. I will not take the time to note each line number this occurs on, but just ask the authors to correct the in-text citations.

Line 35: word choice "typically cover" – change to "typically supply"?

Line 44: Add Burriesci et al. 2012 citation? Citing glucose as the major sugar transported from algae to host?

Line 62: "Bacteria" – broaden this to the microbiome? Inclusive of archaea, fungi, viruses, etc.?

Line 78: Reference citation needed

Line 81: can the authors provide some examples of "the specificity and selectivity of the symbiotic relationship"?

Line 113: Add "(Stanford University)" after Pringle Lab?

Line 208: it might be useful to draw circles around each group/population on the PCA plots to show the distinction and separation

Lines 252-253: I think this repression of apoptosis has been seen in Australian *Aiptasia* studies as well. Check Ashley Dugan's work and potentially cite that here if relevant.

Line 314: Reference citation needed

Lines 355-358: This is where the authors make the conclusion that CC7 is more thermally tolerant than H2. This can't really be said without looking at CC7-SSB01 and H2-SSA01 compared to CC7 and H2. We can't ignore that they harbor two different types of algae endogenously.

Line 387: change "himself" to "itself"?

Reviewer #3 (Remarks to the Author):

This manuscript by Cziesielski and colleagues uses transcriptomics to assess the effect of elevated temperatures on gene expression in two different *Exaiptasia* genotypes and one of those genotypes when colonized by a heterologous dinoflagellate symbiont. The work is straightforward, clearly written, and the results interesting. I think it can make a valuable contribution to the literature after major revisions. I have one major concern about the interpretation of the data, which is central to some of the arguments made, as well as some questions about methods and a few more minor comments.

Most importantly, I disagree with the data interpretation on lines 20, 246 and elsewhere. The authors argue that differences in gene expression and caspase-3 activation between the three symbioses are, at least in part, due to the genotype/physiology of the symbiont. I don't think there is sufficient evidence for that here. The same results could be explained by both lower host thermal resilience of H2 anemones (independent of symbiont genotype) and the effect of heterologous symbiosis in CC7-B01, even if all of the algal symbionts were physiologically the same. While of course there probably are physiological differences between these species, there's no algal physiology presented here. I think the differentiation between *Aiptasia* genotypes under thermal stress and the effect of heterologous symbiosis (as discussed on lines 400-413) is very interesting by itself without this extra interpretation. If the authors have more evidence or references of physiological differences between *S. linuchae* and *B. minutum* (either from their own work or references) that would support their case. The limited physiology data here require the authors to lean heavily on other studies. On a related note, there doesn't appear to be any analysis of DEGs between different symbioses at control temperatures (e.g. CC7 vs CC7-B01 at 25C). While that has been done in several other studies, it would be valuable to present here, while rightly leaving prominence to the thermal stress data.

There is considerable discussion of individual genes within the context of GO categories. That is fine, but the large majority of these gene-level results are not presented as figures, aside from Figs. 5 and 6. This makes the reader rely on a close reading of the text for many of the results. Understandably one can't present over a thousand genes in figure form, but the authors should consider adding some means of presenting up/down regulation of genes discussed in the text. For example, was the NPC result on line 311 up- or down-regulated? In my opinion, if it's worth mentioning in the text, it shouldn't be presented only in supplemental material.

In the methods, more detail on the DEG analysis is needed. I'm not familiar with sleuth, what algorithms/methods are used? What are the FDR and fold-change thresholds required to be considered a DEG? This is very important to state explicitly.

Structurally, the results section contains a lot of material that I would consider data interpretation, and therefore should be in the discussion section. I realize that that can be awkward to write and result in a lot of duplication between the Results and Discussion; consider combining the two if necessary.

Line-by-line comments are below.

92 The current, seemingly ever-changing nomenclature is *Exaiptasia diaphana*, I believe. "Aiptasia" is fine for use in the manuscript (I prefer it myself) but the proper name should be given at least once somewhere.

95 different species, not strains

138 "to mimic natural conditions" This isn't necessary. As stated later in the paper, this experiment is an acute thermal stress condition (heat shock) and has little relationship to the natural environment. While some environments can see rapid temperature shifts like these (shallow reefs, pools, etc.), any animals there would be pre-conditioned to regular (daily?) changes in temperature and would not be comparable to the experiment here. This is absolutely fine and valid biology, just don't call it natural.

177 The symbiont density was calculated by flow cytometer. Were there any differences in symbiont density between symbioses? Was there any indication of bleaching? These results are not presented, as far as I can tell. Also, were there any changes in biomass of the host in response to thermal stress?

198 "A Pearson's correlation analysis between CC7..." Is a difference of 0.01 in r^2 values meaningful?

236 Compare/contrast to proteome-based studies, e.g. Oakley et al. 2017 "Thermal shock induces..." which has some similar results.

265 "Indicated the clade B symbionts shared by H2 and CC7-B01 were enhancing the overall stress experienced by the host" There is no evidence for this, as above. Also, avoid the cladal designations, use *Breviolum*.

287-299 (and 341, 374-399) See my comments above re: effect of symbiont. In general I find heat maps such as Fig. 4 to not be very informative, and I struggle to gain much here. It appears that under heat stress, in each symbiosis, about one third of genes are upregulated, one third downregulated, and one third unchanged. Is this meaningful or distinguishable from a random distribution?

290 I don't know what is meant by "dynamics of symbiosis"

333 "acute heat stress" is an accurate descriptor of the experiment.

378 I think ROS leakage from symbionts to hosts is contested, but see Rehman et al. 2016 *New Phytologist* "Symbiodinium sp. Cells produce..."

405-407 Also see Sproles et al. 2020 *Env. Microbiology* "Sub-cellular imaging shows reduced photosynthetic carbon..."

410 "Symbiont densities were equivalent" Is that in this manuscript or Radecker et al?

424 "the observed reduction in assimilation rates..." You haven't observed any assimilation rates, this is a transcript study, please re-phrase.

Dear Editor,

We would like to thank you and the 3 reviewers for evaluating our manuscript and for providing useful comments that allowed us to improve our study. Please find below detailed responses to the reviewer comments, highlighting the changes we have made to the manuscript to address them.

Reviewer #1 (Remarks to the Author):

This manuscript broadly addresses the effects of thermal stress on transcription and caspase activation in the cnidarian-dinoflagellate symbiosis using the model symbiotic sea anemone *Aiptasia*. Specifically, it examines the interaction between host and symbiont genotype under heat stress. Importantly, it compares the profiles of sea anemones and their homologous symbiont species to a commonly used heterologous combination of CC7 and the algal strain SSB01 (*Breviolum minutum*). This pairing of CC7 and this heterologous symbiont has been used in many studies as a tool to model a healthy symbiosis state, indeed the standard, for the transcriptomic study of symbiotic conditions in *Aiptasia*. However, in this study, the authors find that this heterologous association displays transcriptomic hallmarks of metabolic incompatibility. Together with transcriptomic data, the authors also further examine apoptotic response to heat stress using a caspase assay and show activation in heat-susceptible combination of H2-B. *minutum* and CC7-B. *minutum*, while no response is found in CC7-S. *linucheae*. Finally, the differential expression from the heat stress dataset is viewed in the context of a set of upregulated/downregulated symbiotic state genes generated by a previous meta-analysis.

This work is composed of a straightforward analysis of RNA-Seq data, coupled with a follow-up caspase-like activity assay and a comparison to a previous study that generated a symbiotic state gene list. Overall, the paper is well-written and clearly understood. Methods are clear but brief and figures communicate findings well. Comparing the transcriptomes of homologous combinations to heterologous combinations of *Aiptasia* under heat stress is novel and of broad importance to the coral symbiosis field, with implications for past and future studies that use *Aiptasia* and specifically CC7-SSB01 partners as a symbiosis model. The work accompanying the transcriptomic dataset fits into the story, but the figure presentation occasionally feels incomplete. More transparency and detail is needed in general to allow for reproducibility.

In conclusion, this manuscript will help the coral symbiosis field understand how host genotype and symbiont identity influence response to heat stress, two areas of research that are critical to understand in the face of coral decline due to climate change. The conclusions drawn from this paper mesh well with previous studies of heterologous associations in *Aiptasia* and extend evidence of symbiotic incompatibility into thermal stress conditions.

I have several general points as well as some specific notes on lines in the text.

Comment 1: The use of homologous and heterologous should be defined and justified early in this paper. Often heterologous is used to describe partners that are non-native to their host species. In *Aiptasia* this has often been with species such as *D. trenchii* or *S. microadriaticum* which are never found in the host in nature. However, here both species *B. minutum* and *S. linucheae* are found in naturally occurring populations of *Aiptasia*, they just differ based on geographical location.

We have inserted the following text in line 86 to help clarify the definitions and use of the terminology.

‘We refer to this strain as heterologous in this study, reflecting the fact that while it may be natural for *Aiptasia* to host SSB01 symbionts, the genotype CC7 is not commonly associated with this symbiont strain. As such, we refer to CC7-B01 as a heterologous combination to reflect a non-native symbiosis, while CC7 and H2 reflect the geographically native host-symbiont pairings and are therefore here considered as homologous’

Comment 2: With respect to the methods, I found myself needing clarification in several places, which I will separate out for ease of addressing:

2a: Paragraph at Line 123: In this study the authors compare sea anemones with their homologous symbiont to sea anemones that were bleached and recolonized with SSB01. Was there evidence that all three combinations had roughly the same symbiont density and were at the same stage of symbiosis (e.g. not still in the process of recolonization)? Is there any visual or cell counting information to show that algal densities were roughly similar between all three combinations, since one originated from aposymbiotic animals? This would be useful information to have, as other studies of *Aiptasia* had heterologous colonization for one year and found much lower symbiont density in the heterologous pairing (Medrano et al., 2019). If no data is available, there at least is some reference to equivalent densities buried in the discussion at line 410. At the very least I would expand on this statement in this method paragraph.

We thank the reviewer for pointing this out and we have now added the symbiont count data as Table 1, and we can confirm that all host symbiont combinations were fully symbiotic. Regarding CC7-B01, while symbiont counts for the specific experiments presented here do show differences between CC7 and CC7-B01, with the heterologous CC7-B01 combination showing higher average symbiont counts at 25C, these differences were not significant anymore at 32C.

Furthermore, we would like to point out that CC7-B01 has a general tendency to show higher symbiont counts when compared to CC7 and we would like to refer to one of our previous studies. Please see figure S5 in Herrera et al. 2020 where we show that symbiont densities were higher in CC7-B01 compared to the native CC7-A4 combination, albeit nonsignificant. We’re now also referring to this study to provide the readers with this essential information.

M. Herrera, S.G. Klein, S. Schmidt-Roach, S. Campana, M.J. Czielski, J.E. Chen, C.M. Duarte, M. Aranda. “Unfamiliar partnerships limit cnidarian holobiont acclimation to warming”. *Global Change Biology* 2020; 00: 1– 15. <https://doi.org/10.1111/gcb.15263>

2b. Paragraph at line 151: please include accompanying scripts in the supplemental data or at another specified repository for best practices of data reproducibility.

Thank you for the comment. We have now made our code and intermediate files available at https://github.com/lyijin/increased_incomptability to ensure reproducibility of our analyses.

We have mentioned this in Line 753 of the manuscript.
“Code and intermediate files are available at https://github.com/lyijin/increased_incompatibility.”

2c. Paragraph at line 165: Please include the sample size for this experiment. If sample number is the same as the previous experiment be more explicit. Also, this paragraph indicates Symbiodiniaceae were counted at line 177, but I can’t find the corresponding symbiont count data in the results section. This would be useful to see any differences in symbiont density. In addition, the volumes of lysed homogenate used for the caspase assay and for the protein assay should be included. It should further be noted in the methods that this assay measures caspase 3-like activity, including caspase-3 and other DEVD-specific activity.

Thank you for these remarks. We have addressed them in the following:

- We have added a brief repetition of the procedure, highlighting the number of replicates used and their set up. Line 153 reads: ‘A follow-up experiment with the same procedures as before was conducted to test for caspase-3 activation: twelve anemones for each strain were taken from two separate tanks, with four individuals used per treatment.’
- We have renamed the results section ‘Associated symbionts influence Caspase-3 mediated apoptosis’ to ‘Physiological response of host-symbiont combinations. In this section, we now feature the results from caspase activity as previously and have also added our data (which was indeed previously missing from the supplementary materials and we apologize for that) on symbiont count densities. The section now reads (line 277):

“Since symbiont density plays a critical role in the stress experience of the host, and could influence observed caspase-3 activities, we counted *Symbiodiniaceae* cells per replicate and normalized the counts against total protein. Table 1 shows the normalized symbiont density, which showed no significant difference between the two treatments. Note that symbiont counts could not be conducted on the same individual before and after incubation time, due to the method requiring the anemone’s homogenate. As such, the data shown here indicates that there was no significant difference between animals in the different treatment condition at the end of the incubation period.”

Table 1. – Normalized symbiont density (n=4 per strain-temperature combination). Units are counts [ml filtrate]⁻¹[μg Aiptasia protein]⁻¹. No significant changes in symbiont counts were detected between treatments (two-tailed t-test).

Strains	25 Degrees				32 Degrees				p value
	rep1	rep2	rep3	rep4	rep1	rep 2	rep 3	rep4	
CC7	13,435	10,703	10,314	10,994	17,107	20,738	10,152	28,091	0.091
H2	17,327	11,896	9,987	9,090	14,777	14,567	1,197	13,656	0.424
CC7-B01	27,811	17,044	28,412	18,159	11,808	26,015	10,951	24,080	0.34

Information regarding the volume of lysed homogenate used is mentioned in line 163, where we now clarify that 100 μl was used for the Caspase-3 Assay, and 200 μl for the Micro BCA Protein Assay kit, as well as 25 μl homogenate for the Guava flow cell count. We have added a sentence in line 161 mentioning the caspase-3 and DEVD specific activity.

Comment 3: I have a general question regarding whether differences seen between CC7 and H2 are biological as stated or if there are differences related to technical processing:

3a: For example, it would be useful to have a supplementary table showing specific information for CC7/H2/CC7-B01 on read counts, percentages of successfully mapped reads, and resulting gene model coverages. Or simply include text in the results e.g. at line 185 that briefly states read counts, mapping and gene model coverage were similar between pairings. The reason why I am concerned is the presentation of Figure 4/Table S3. Figure 4 has me concerned that transcriptome profiles for H2 samples are narrower/shallower than CC7 samples. If so this could mean that the lower amount of DE genes from H2 could be partially attributed to technical difference rather than biological difference. Since reads are mapped to Aiptasia gene models created from CC7 genotype, evidence or lack of evidence for this mapping bias would be very useful to know with regards to Figure 1, 2, and 4.

To address the concern of the reviewer right away, no, the number of mapped reads was not lower in H2, and neither were the number of identified expressed genes in H2. In fact, the sample with the lowest number of mapped reads was a CC7 sample with >6 million mapped reads while the lowest H2 sample had >8 million mapped reads. To provide more detailed information on the stats etc. we now included an excel sheet (expr_stats.xlsx) with the number of sequenced and mapped reads for every sample on the project github page. https://github.com/lyijin/increased_incomptability/blob/master/differential_gene_expression/expr_stats.xlsx

3b. Figure 1 is clearly presented and described. The authors explain how genotypes clearly differ, and how CC7 genotype changes significantly with respect temperature stress. H2 genotype does not cluster separately but it is unclear whether that difference comes with the caveat of restricted overall transcriptomic profiles.

As mentioned in our response above, there are no big differences in the number of mapped reads or the number of expressed genes between CC7, H2 and CC7-B01. Furthermore, H2

samples had an average of 10.7 million mapped reads, with the lowest sample having >8 million mapped reads, while CC7 samples had an average of 8.5 million reads, with the lowest sample having >6 million mapped reads. Based on these numbers, we are confident that the results presented here are not biased by restricted transcriptomic profiles.

Comment 4: Figure 2 is also clearly presented. Since there is no mention of mRNA splicing yet it is 25% of the figure, I think readers would benefit from the inclusion of a short blurb about its continued presence in coral symbiosis heat stress studies similar to the paragraph found in Cziesielski et al. 2018 <http://dx.doi.org/10.1098/rspb.2017.2654>.

Thank you for your feedback. We have now included a few lines discussing this observation briefly in the discussion section, starting line 359 reads: 'Notably, the results indicate that splicing related genes are differentially regulated across the strains investigated. Splicing repression can act as a form of protein production control in eukaryotic cells that allows for the selective expression of proteins in response to stress⁵², and the expression of related genes has previously been reported in coral heat stress studies^{46,53}.

Comment 5: Figure 3 should have proper units on the y-axis, with μmol of product (R-110) per unit time (h) per unit of protein (mg). If no reference standard was used, then signify arbitrary units in the legend. Currently it appears as if fluorescence is normalized to total protein per animal, but it would make more sense to normalize by the amount of protein present in the equivalent volume used for the caspase assay. Was the inhibitor from the kit used as a control to verify that the fluorescence signal is specific to caspase? If so, I would include that data.

Thank you for these notes. Indeed, we did normalize the fluorescence against the amount of protein present in the equivalent volume used for the caspase assay. As previously stated in the methods, we used 100ul for caspase, and 200ul for the protein assay. In our calculations, we took the total protein volume and divided it by 2 to normalize to the caspase volume used. We now also provide all this data and our analysis in the supplementary files (Table S4). We also refer to it in the text in line 274).

The figure has been amended for the y-axis to read 'Caspase activity (RFU/mg of protein)' and since we did not use a reference value, we made sure to present this in the legend by adding the line (line 275): 'Caspase activity was measured in relative fluorescent units and normalized per unit of protein (mg).'

We did not use an inhibitor for verification and are therefore unable to present this data.

Comment 6: Figure 4 is very interesting but difficult to read. It would benefit from having clear titles for the separate upregulated and downregulated sections. The column labels are fine in the middle, but the figure would benefit from labels being placed on the top as well. The legend should make clear the organizational premise of the figure, that column E is a gene list from a previous study identified as upregulated or downregulated during symbiosis. The legend should also indicate how significant fold-change was determined.

Thank you for the suggested changes. We have expanded on the caption of the figure to outline the provenances of each column and the meaning of the indicated values more clearly. In the figure, we have added labels to explain the expected changes in gene expression (at the side however, not at the top as suggested).

Comment 7: I have a bit of confusion with the underlying data of Figure 4 and Table S3.

Comment 7a: As its written in Figure 4, “clear columns represent no significant fold change between treatments”. In the accompanying supplemental data Table S3, a majority of the genes show a fold change of exactly “0”. Does this mean they were below the significance cutoff? If so this should be stated in Table S3. However, at the top of Table S3 the qualification is “at least one strain has to have the gene expressed to be considered”. Does this mean the gene was not present in that sample’s transcriptome? If so, it needs to be clarified in Figure 4 and Table S3 that white areas or “0”s indicate absent data, and not true measurements of fold change.

We apologise for not fully clarifying how the values in Fig 4 and Table S3 were calculated.

Fold change values in Fig 4 and Table S3 are log₂-transformed. As you correctly surmised, non-zero values represent genes with significant differential expression ($p < 0.05$); while zeroed values represent genes with non-significant differential expression ($p > 0.05$).

Zero here does not mean “no data”, it means “not significant”. If a gene had non-significant fold changes in all three strains, they were not plotted (hence the disclaimer that at least one of the three strains must have a non-zero value for the gene to be represented in Fig 4 and Table S3).

Comment 7b: It would be useful for Table S3 to include gene annotations, so readers don’t have to switch over to Table S1 to gain information. As well as the significance cutoff or cutoffs used to categorize genes as significantly upregulated or downregulated.

The meaning of the tables has been addressed as per response in the previous comment and we have now included gene annotations in table S3.

Comment 8: (Line 320) Regarding the Figure 5 legend: the wording of the parenthetical is ambiguous. Was the expectation to be upregulated in heat stress or in symbiotic state? Overall, I think the figure and the rest of the text does a good job showing that ammonium assimilation is reduced more under thermal stress in heterologous combination.

The legend now reads: ‘The most significant response changes were observed in CC7-B01, which expressed the highest number of symbiosis genes in the opposite direction from the expected.

Comment 9: Regarding Figure 6, the callback to the Cui et al. 2019 figure is well done and helps show exactly how heat stress alters these two pathways during symbiosis.

Thank you very much!

Comment 10: Regarding CC7 and its geographic location. CC7 originates from an Aiptasia population sent to the Pringle lab from a North Carolina company (Carolina Biological) (Sunagawa et al. 2009). This is different than being collected on the North Carolina coast, which is much more temperate and where Aiptasia contains *B. psymophilum*, a cold-tolerant symbiont found along the eastern coast of the United States. *S. linucheae* is thermally tolerant to heat but has only been collected from the Caribbean previously (Diaz-Almeyda et al

2017, https://www.algaebase.org/search/species/detail/?tc=accept&species_id=145014).

To my knowledge *S. linucheae* samples have not been collected off of North Carolina.

However, it is commonly found as a symbiont to *Exaiptasia* found in Florida (Medrano et al. 2019). This corresponds with CB website <https://www.carolina.com/marine-and-saltwater-animals/sea-anemone-aiptasia-living/162865.pr> which says they ship anemones directly from a vendor on the

Florida coast. All this to say, because of CC7 unknown origins, it is difficult to say how thermally distinct the original locations of the anemone clonal lines are. This is important for the rest of the paper especially when attempting to compare relative thermal regimes of locations. CC7 is a fantastic tool to study symbiosis but the strain should not have ecological meaning thrust upon it. This location name propagates through studies and misinforms readers. Below I've included lines where reference to this geographical location may need to be altered or clarified.

We totally agree with the reviewer, but this issue is a problem that is difficult to address. In previous studies, we did indeed refer to this strain as originating from Florida, but reviewers insisted that we have to refer to it as originating from North Carolina. Having said that, analysis of the SCAR markers previously published by Thornhill et al. 2013 shows a specific 13 bp insertion in marker SCAR5 that is only found in Aiptasia samples from Florida, and we do find the same deletion in CC7. This, together with the fact that CC7 is natively associated with *S. linucheae* which according to Thornhill et al 2013 is also exclusively found in Aiptasia from Florida, makes us confident in assuming that the CC7 strain indeed originates from Florida. We have now changed this in the manuscript and hope that another reviewer doesn't object.

Line 16: "two thermally distinct locations, North Carolina (CC7) and Hawaii (H2)."

Line 95: "distinct locations, North Carolina (CC7) and Hawaii (H2),"

Line 112: "CC7 originated from North Carolina"

Line 473: "due to its natural habitat in North Carolina"

Comment 11: The discussion of this work and its comparison with some previous homologous v heterologous studies in Aiptasia and other species is lacking. For example, Matthews et al. 2017 (10.1073/pnas.1710733114) shows a similar heterologous transcriptomic shift in "stable symbiosis", and Medrano et al. 2019, which uses CC7 and *D. trenchii* as partners, shows similar patterns in its protein dataset

(10.3389/fmicb.2019.01153). Lin et al. 2019 also shows transcriptional differences between homologous/heterologous symbionts with a corallimorpharian (doi.org/10.1242/bio.038281).

We agree with the reviewer that we did not discuss our results considering the literature mentioned, mostly because in contrast to the mentioned studies, we used symbiont strains that are naturally associated with *Aiptasia*. We have now added a few sentences to put our results into perspective to the studies mentioned. Specifically, we added the following:

This is reminiscent of findings in studies using heterologous symbionts in *Aiptasia* and the corallimorpharian *Ricordea yuma*, which observed transcriptomic responses intermediate between symbiotic and aposymbiotic individuals, and suboptimal nutrient exchange (Matthews et al. 2017, Lin et al. 2019).

We also added a reference to Medrano et al. 2019 to the following sentence:

The fact that CC7-B01 carries the same symbiont as H2, which is the homologous host of the B01 symbiont, therefore, further suggests that successful and resilient symbiosis relies on the physiological compatibility of host and symbiont and echoes previous findings in *Aiptasia* ³⁶(Medrano et al. 2019).

Comment 12: Bibliography has several errors that require correcting and that result in numerous in-text citation problems (e.g., at lines 32, 55, 80, 379, 384).

We have adapted the referencing style to Comms. Bio's standard referencing style (mirroring that of Nature), and fixed references in the bibliography.

Minor comments:

Line 92: The species name of *Aiptasia* is never used in the manuscript and should at least occur here in parentheses as "*Aiptasia* (*Exaiptasia diaphana*)".

Applied

Line 96: fix spelling of "Symbiodinium linucheae"

Applied

Line 143: Can the authors provide an estimate of animal size (e.g. oral diameter)? There is currently no reference here for what "larger" entails.

The selection of 'larger' animals was based on visual experience, rather than actual measurements. A rough estimate is that animals had to have at least 0.5 cm in height when rested. However, since we do not have measurement data for this, we have opted to remove the sentence and only state that the animals were of similar size.

Line 158: fix spelling of “duplicate”

Applied

Lines 160: Please add version information to sleuth. As well as significance cutoffs for differential expression analyses.

Provided

Line 162: Please add version information for topGO and specify what default generally entails.

Provided

Line 171: fix spelling of “EnzChek”

Applied

Line 375: Symbiont species can differ in temperature resilience.

Applied

Line 387: change “himself” to “itself”

Applied

Line 397: I don’t know what “exasperated” means in this context.

After careful reading we decided that the sentence is not required as the previous and following sentences are sufficient to make the statement we intended.

Reviewer #2 (Remarks to the Author):

This study by Cziesielski and colleagues investigates the role of both host and algal symbiont in thermal tolerance of the cnidarian holobiont. The authors use the fact that *Aiptasia* is compatible with multiple strains of symbiotic algae and constructed a heterologous combination between CC7 anemones and SSB01 (derived from H2 anemones) and then used RNA-sequencing to compare the transcriptional profiles of each strain under control and thermal-stress conditions. The study found that CC7 and CC7-SSB01 has similar transcriptional profiles and that CC7 and H2 remained distinct throughout the thermal stress experiment. While symbiosis-associated genes were differentially expressed in all strains in response to heat, the CC7-SSB01 combination showed a more dramatic response, potentially suggesting that the heterologous nature of that symbiosis was less stable than those of CC7 and H2 with their endogenous symbionts.

This study is both important and timely in order to better understand the dynamics of symbiosis overall in the cnidarian-dinoflagellate system and especially under thermal-stress.

The manuscript is well written and easy to follow and read. The conclusions reflect the data generated from the experimental design (with the noted exception below). The statistics and bioinformatics are justified and explained clearly with appropriate citations.

The work provided here is convincing based on the data presented. However, the one addition that could have made this comparison across strains stronger would be to have the reciprocal heterologous symbiosis with H2 anemone with SSA01 algae derived from CC7 anemones. It is a severe limitation of this study to only have CC7 with the heterologous algal strain. The authors make conclusions that CC7 may be the more thermally-tolerant anemone due to the results of CC7-SSB01 and compared to H2 but of course, the difference between CC7 and H2 is not just the anemone but also the algal species harbored endogenously by both anemones. This study still provides useful insights into the dynamics of symbiosis and especially under thermal stress. To be more complete, either more strains of anemones or more heterologous combinations could have been included. This is not a fatal flaw but this type of comparison needs to be done to really tease out the role of both host and algal symbiont in thermal tolerance of the holobiont.

We agree with the reviewer, but unfortunately, we did not have the isolated *S. linucheae* strain from CC7 available at the time these experiments were performed. However, as the reviewer states, we also believe that this is not a fatal flaw, but a full factorial experimental design should definitely be considered for future studies.

The RNASeq addition here provides a nice snapshot at gene expression during thermal stress, however a time series approach would have also identified genes that are differentially expressed in the strain combinations before, during, and after bleaching. As it is, there is no clear mention in the text when the anemones were sampled for RNA extraction after being exposed to heat stress. “The heat stress duration (24 hours) lasted from noon until noon the next day”. So, anemones were sampled immediately after? We know from Cleves et al. 2020 (PNAS) that there are strongly differentially expressed genes very soon into the heat stress and return to baseline well before bleaching actually occurs. So, picking one time point and making lots of conclusions based on genes that are differentially expressed at that point is difficult.

We agree with the reviewer but a time series with multiple host symbiont combinations increases the number of replicates, and the downstream costs for sequencing etc. dramatically. In the future, similar time series as performed by Cleves et al (but using only CC7), would surely provide a more refined view of the stress response. However, we would also like to point out that we independently tested our finding of increased caspase activation, which provides independent functional validation of at least one the conclusions we made from our observed differences in gene expression.

More detailed/specific comments:

Throughout the draft, there are numerous inconsistencies with the in-text citations. In many cases, the author list needs to be truncated to “et al.” after the first author. I think this is a result of the citation management software, but this needs to be corrected. I will not take

the time to note each line number this occurs on, but just ask the authors to correct the in-text citations.

We have adapted the in-text referencing style to that of Nature, as recommended by Communications Biology in their submission guidelines. The in-text situations are therefore numerical now and the mentioned inconsistencies have been fixed.

Line 35: word choice “typically cover” – change to “typically supply”?

Line 44: Add Burriesci et al. 2012 citation? Citing glucose as the major sugar transported from algae to host?

Line 62: “Bacteria” – broaden this to the microbiome? Inclusive of archaea, fungi, viruses, etc.?

All above applied

Line 78: Reference citation needed

We discuss the relevant references that support our statement of “However, inoculation with heterologous thermotolerant symbionts does not necessarily improve temperature resilience of the host.” In the remaining parts of this paragraph we include findings from studies such as Herrera et al. 2020, LaJeunesse et al., 2004 and Parkinson and Baums, 2014.

Line 81: can the authors provide some examples of “the specificity and selectivity of the symbiotic relationship”?

We’re not sure if we're supposed to mention specific examples as the statements refer to the general issue of partner specificity in cnidarians and we already provide references to specific studies where this is discussed in detail. For instance, we refer to our previous work in Herrera et al. 2020 on specificity vs environmental conditions. In case this is deemed essential we're of course willing to extent our introduction to include more background on this topic, but we would need a bit more specific information on what exactly.

Line 113: Add “(Stanford University)” after Pringle Lab?

Added

Line 208: it might be useful to draw circles around each group/population on the PCA plots to show the distinction and separation

The primary purpose of the plot was to show the clear split in CC7’s versus H2’s stress response. We believe plot A does that, assisted by the color spectrum used (i.e. distinct blue for H2).

Lines 252-253: I think this repression of apoptosis has been seen in Australian Aiptasia studies as well. Check Ashley Dugan’s work and potentially cite that here if relevant

Unfortunately, we could not find this study using the provided information. If referencing this study is deemed essential, we would welcome a complete reference with authors, title, year and journal.

Line 314: Reference citation needed

We included Cui et al., 2019 here.

Lines 355-358: This is where the authors make the conclusion that CC7 is more thermally tolerant than H2. This can't really be said without looking at CC7-SSB01 and H2-SSA01 compared to CC7 and H2. We can't ignore that they harbor two different types of algae endogenously.

Indeed, this is true, and we did report the observations as 'suggested', in line with evidence from other studies. However, to acknowledge this issue further we have edited the sentence to read as following

Line 370: 'While we recognize that endogenously H2 and CC7 are associated to different types of symbionts, the up-regulation of pathways crucial to an efficient stress response in CC7 suggests that this genotype has a superior thermal response compared to H2...'

Line 387: change "himself" to "itself"?

Applied

Reviewer #3 (Remarks to the Author):

This manuscript by Cziesielski and colleagues uses transcriptomics to assess the effect of elevated temperatures on gene expression in two different *Exaiptasia* genotypes and one of those genotypes when colonized by a heterologous dinoflagellate symbiont. The work is straightforward, clearly written, and the results interesting. I think it can make a valuable contribution to the literature after major revisions. I have one major concern about the interpretation of the data, which is central to some of the arguments made, as well as some questions about methods and a few more minor comments.

Most importantly, I disagree with the data interpretation on lines 20, 246 and elsewhere. The authors argue that differences in gene expression and caspase-3 activation between the three symbioses are, at least in part, due to the genotype/physiology of the symbiont. I don't think there is sufficient evidence for that here. The same results could be explained by both lower host thermal resilience of H2 anemones (independent of symbiont genotype) and the effect of heterologous symbiosis in CC7-B01, even if all of the algal symbionts were physiologically the same. While of course there probably are physiological differences between these species, there's no algal physiology presented here. I think the differentiation between *Aiptasia* genotypes under thermal stress and the effect of heterologous symbiosis (as discussed on lines 400-413) is very interesting by itself without this extra interpretation. If the authors have more evidence or references of physiological differences between *S. linuchae* and *B. minutum* (either from their own work or references) that would support their case.

We do not agree with the reviewer as our data shows clear differences between CC7 and CC7-B01 anemones. While the reviewer highlights the heterologous nature of CC7-B01 as a potential source for the observed differences, we argue that this also reflects different physiologies between the two symbiont strains. Regarding the specific statements criticized by the reviewer, in line 19 we state:

“We find that oxidative stress and apoptosis responses are strongly influenced by symbiont type”

This statement is accurate as we do observe this and it doesn't insinuate in any way that these observed differences are based on differences in symbiont physiologies, it merely states the differences we observed in host response.

In line 235 we state, “The enriched oxidative stress response in the two strains sharing the same symbionts, CC7-B01 and H2, indicated that the symbiont might be exerting higher cellular stress on these hosts”. Again, this statement doesn't specifically state if this is a response resulting from physiological differences in the symbiont or potential host symbiont interactions. It merely reflects our observation that host-symbiont combinations with this specific strain show higher oxidative stress and apoptosis responses. Furthermore, it could be argued that any differences in host-symbiont interactions should ultimately also be grounded, at least in part, on physiological differences between the different symbionts.

Having said that, we do have independent data showing that symbiont physiologies are indeed different, and the manuscript is currently being drafted. Furthermore, we have recently shown that the differences observed between different host-symbiont pairings seem to be predominantly dictated by the host (Herrera et al. 2020) as we also state here. However, since we do not want to include this data in this study, we refrain from mentioning it here.

The limited physiology data here require the authors to lean heavily on other studies.

We would like to point out that we already present more physiological data than most studies in the field. For example, see:

Cleves PA, Krediet CJ, Lehnert EM, Onishi M, Pringle JR. Insights into coral bleaching under heat stress from analysis of gene expression in a sea anemone model system. *Proc Natl Acad Sci U S A*. 2020 Nov 17;117(46):28906-28917. doi: 10.1073/pnas.2015737117

Mohamed AR, Andrade N, Moya A, Chan CX, Negri AP, Bourne DG, Ying H, Ball EE, Miller DJ. Dual RNA-sequencing analyses of a coral and its native symbiont during the establishment of symbiosis. *Mol Ecol*. 2020 Oct;29(20):3921-3937. doi: 10.1111/mec.15612

Medrano E, Merselis DG, Bellantuono AJ, Rodriguez-Lanetty M. Proteomic Basis of Symbiosis: A Heterologous Partner Fails to Duplicate Homologous Colonization in a Novel Cnidarian- Symbiodiniaceae Mutualism. *Front Microbiol*. 2019 May 31;10:1153. doi: 10.3389/fmicb.2019.01153

Lin MF, Takahashi S, Forêt S, Davy SK, Miller DJ. Transcriptomic analyses highlight the likely metabolic consequences of colonization of a cnidarian host by native or non-native Symbiodinium species. *Biol Open*. 2019 Mar 27;8(3):bio038281. doi: 10.1242/bio.038281

On a related note, there doesn't appear to be any analysis of DEGs between different symbioses at control temperatures (e.g. CC7 vs CC7-B01 at 25C). While that has been done in several other studies, it would be valuable to present here, while rightly leaving prominence to the thermal stress data.

We do agree that comparing 25C treatments to each other could provide interesting insights but it would also increase the amount of information and distract from the main findings while increasing the word count beyond the limit for Articles. In it's current state, the manuscript is just at the word count limit. Furthermore, such a comparison would require comparing treatments between genotypes, i.e. between CC7 and H2, which could introduce biases as the reference transcriptome used was from CC7 only. This is not an issue in the current study as we only compare transcriptional differences between treatments in the same genotype.

There is considerable discussion of individual genes within the context of GO categories. That is fine, but the large majority of these gene-level results are not presented as figures, aside from Figs. 5 and 6. This makes the reader rely on a close reading of the text for many of the results. Understandably one can't present over a thousand genes in figure form, but the authors should consider adding some means of presenting up/down regulation of genes discussed in the text. For example, was the NPC result on line 311 up- or down-regulated? In my opinion, if it's worth mentioning in the text, it shouldn't be presented only in supplemental material.

In response to the reviewer's comment we have now included the gene IDs and the fold changes of the respective genes mentioned in the main text. We have further added the annotations to table S3 so that changes in expression for the different genes and host symbiont combinations can be looked up easier. The changed text now reads: "However, CC7-B01 stood apart from the homologous host-symbiont combinations by several genes relating to critical symbiosis nutrient exchange pathways: aldehyde dehydrogenase family members (AIPGENE12723: fc -0.54; AIPGENE19640: fc -0.48; AIPGENE21619: fc -0.76), Niemann-Pick disease protein (NPC) (AIPGENE5532: fc -0.501), glutamine synthetase (AIPGENE26763: fc -0.28), glutamate dehydrogenase (AIPGENE26078: fc -0.33), and ammonium transporters (AIPGENE17420: fc -0.52 ;AIPGENE18105: fc -1.27) were down-regulated in CC7-B01."

And:

"Genes involved in amino acid synthesis, such as betaine-homocysteine S-methyltransferase (BHMT: AIPGENE10977: 1.03), dimethylglycine dehydrogenase (DMGDH: AIPGENE10986: fc 1.4; AIPGENE10961: fc 1.35) and 4-hydroxyphenylpyruvate dioxygenase (HPD: AIPGENE25576: fc 0.65) also displayed expression changes in the opposite direction compared to stable symbiosis in CC7-B01. Indeed, during stable symbiosis these genes are

expected to be down-regulated and/or with no significant change in expression as was the case for H2 and CC7.”

In the methods, more detail on the DEG analysis is needed. I’m not familiar with sleuth, what algorithms/methods are used? What are the FDR and fold-change thresholds required to be considered a DEG? This is very important to state explicitly.

Thank you for pointing this out. We have now added the information to the material and methods section to show the statistical methods and cut-offs used in our differential gene expression analysis.

Structurally, the results section contains a lot of material that I would consider data interpretation, and therefore should be in the discussion section. I realize that that can be awkward to write and result in a lot of duplication between the Results and Discussion; consider combining the two if necessary.

We understand the reviewer’s suggestion but given the amount of work and likely re-assessment of the manuscript if the structure is changed to combine Results and Discussion we prefer to maintain the structure. Further, we believe that the changes we made already address some of these issues.

Line-by-line comments are below.

92 The current, seemingly ever-changing nomenclature is *Exaiptasia diaphana*, I believe. “Aiptasia” is fine for use in the manuscript (I prefer it myself) but the proper name should be given at least once somewhere.

We now mention the name in line 77:

“To do this, we used the sea anemone *Aiptasia (Exaiptasia diaphana)*, which has emerged as a flexible model system to study cnidarian-algae symbioses.”

95 different species, not strains

Adapted

138 “to mimic natural conditions” This isn’t necessary. As stated later in the paper, this experiment is an acute thermal stress condition (heat shock) and has little relationship to the natural environment. While some environments can see rapid temperature shifts like these (shallow reefs, pools, etc.), any animals there would be pre-conditioned to regular (daily?) changes in temperature and would not be comparable to the experiment here. This is absolutely fine and valid biology, just don’t call it natural.

Thank you for pointing this out. We have removed this part of the sentence.

177 The symbiont density was calculated by flow cytometer. Were there any differences in symbiont density between symbioses? Was there any indication of bleaching? These results

are not presented, as far as I can tell. Also, were there any changes in biomass of the host in response to thermal stress?

Thank you for this comment which echoes similar statements from reviewer 1. We have now addressed this issue in line 277 – 284 and we now state that the data was obtained from flow cytometry normalized to protein content and that no significant changes in symbiont density was detected between treatments. We now also provide symbiont densities and p-values for treatment comparisons in a new Table 1.

198 “A Pearson’s correlation analysis between CC7...” Is a difference of 0.01 in r^2 values meaningful?

We have amended the sentence to now read (line 187): “A Pearson’s correlation analysis between CC7-B01 and CC7 at 32 °C ($p < 0.001$; $r^2 = 0.96$) and 25 °C ($p < 0.001$, $r^2 = 0.95$) showed a similarly high correlation maintained across conditions.”

236 Compare/contrast to proteome-based studies, e.g. Oakley et al. 2017 “Thermal shock induces...” which has some similar results.

The study by Oakley et al, is based on proteome data, which we do not assess in this study, and we have previously shown in Cziesielski et al. 2018 that there can be discrepancies between proteome and transcriptome. Furthermore, the referred section is in the results, where we tried to avoid going into deeper discussions. However, findings in Oakley et al. 2017 match our observations here and we have therefore referenced the paper in our discussion section. Line 375 now reads: ‘Recent studies have shown that heat shock can induce antioxidant mechanisms and production of other proteins aiding in stabilizing nascent proteins, which is also indicative of endoplasmic reticulum stress⁵⁵

265 “Indicated the clade B symbionts shared by H2 and CC7-B01 were enhancing the overall stress experienced by the host” There is no evidence for this, as above. Also, avoid the cladal designations, use *Breviolum*.

Thanks for pointing this out. We have checked the text and ensured that we continuously refer to *Breviolum* as *B. minutum* rather than *clade B*.

Further, we have tried to phrase the findings less absolute by rephrasing the sentence to: ‘Overall, the GO-term and gene-specific observations indicated the *B. minutum* symbionts shared by H2 and CC7-B01 may have been enhancing the overall stress experienced by the host.’

287-299 (and 341, 374-399) See my comments above re: effect of symbiont. In general I find heat maps such as Fig. 4 to not be very informative, and I struggle to gain much here. It appears that under heat stress, in each symbiosis, about one third of genes are upregulated, one third downregulated, and one third unchanged. Is this meaningful or distinguishable from a random distribution?

We apologize for the confusion. We have improved the annotation of Figure 4 which hopefully better clarifies the point of the figure. Briefly, the heatmap provides a graphical view of the numbers provided in the text, i.e., that CC7 and H2 anemone show ~55% of the symbiosis genes being differentially expressed in the opposite direction during stress while CC7-B01 shows 65% of the genes being expressed in the opposite direction. We have now also included a statistical test to make clear that CC7-B01 expresses significantly more symbiosis genes in the opposite direction than CC7 and H2 ($p < 10^{-6}$, chi-square test with Bonferroni correction).

The amended figure should now make it clearer that the genes towards the top of the heatmap are genes upregulated during symbiosis and these should thus appear in red in the different strains under normal conditions. Conversely, the genes towards the bottom are downregulated under normal conditions and are thus expected to be blue in the different strains.

We plotted the heatmap to visually check common patterns of gene expression across the three tested strains—were most of the genes similarly overexpressed across all three strains (red/red/red across row)? Or do CC7 and CC7-B01 share more similarities than with H2 (red/red/not-red, or blue/blue/not-blue)? After the heatmap was plotted, we noticed that CC7-B01 had more discordantly expressed genes than the other two strains, which motivated our search of what those genes are, and what the biological implications are.

The reviewer is correct in saying that numbers and statistical significances are not easily obtained from heatmaps. As mentioned above we have now included a statistical test to show that CC7-B01 have more discordantly expressed genes than the other two strains ($p < 10^{-6}$, chi-square test with Bonferroni correction). In the process of performing the calculations, we realized that the numbers in the main text was referring to an earlier version of the analysis and was not updated when we re-ran the analysis to fix errors. The differences in the numbers were minor and did not affect the conclusions of the original main text (original numbers were 170 / 88 / 78).

Line 296 now reads:

We found that 168 (CC7-B01), 94 (CC7), and 85 (H2) of these genes were significantly differentially expressed in response to heat stress in the different strains (Fig. 4). Additionally, we found that around 55% of symbiosis genes in CC7 and H2 were regulated in the opposite direction to what is expected in stable symbiosis. In contrast to CC7 and H2, CC7-B01 has significantly more genes (65%, $p < 10^{-6}$, chi-square test with Bonferroni correction) showing changes in the opposite direction to what is expected in a stable symbiosis.

The script used to derive these numbers and perform the test has also been uploaded to the GitHub repo.

290 I don't know what is meant by "dynamics of symbiosis"

Modified. Now reads: 'We, therefore, determined if and how symbiosis was affected under heat stress in different combinations CC7, H2, and CC7-B01.'

333 "acute heat stress" is an accurate descriptor of the experiment.

Thank you.

378 I think ROS leakage from symbionts to hosts is contested, but see Rehman et al. 2016 New Phytologist "Symbiodinium sp. Cells produce..."

This is a great reference, thank you. We have added it to the additional references of Howells et al. 2011 and Cuning & Baker, 2013, who show similar results of leakage.

405-407 Also see Sproles et al. 2020 Env. Microbiology "Sub-cellular imaging shows reduced photosynthetic carbon..."

Thank you for this indication. We have now added a sentence to highlight the reference (line 415):

"As observed in other studies, non-native host-symbiont pairings may result in sub-optimal nutrient exchange under normal conditions⁷⁰, which is likely to worsen under thermal stress."

410 "Symbiont densities were equivalent" Is that in this manuscript or Radecker et al?

The text was in reference to Radecker et al, but we also show in our symbiont counts that there is no significant difference between strains at 32C.

424 "the observed reduction in assimilation rates..." You haven't observed any assimilation rates, this is a transcript study, please re-phrase.

Amended. Now reads (line 437): "If symbiont-derived carbon drives host ammonium assimilation, then the observed reduction in genes involved in ammonium assimilation also reflects reduced photosynthate translocation and, therefore, a metabolic decoupling between partners.

REVIEWERS' COMMENTS:

Reviewer #1 (Remarks to the Author):

I thank the authors for providing revisions to this manuscript which do a good job of clearing up most of my concerns regarding transparency and detail. Though the analysis of the performed experiments are largely restricted to focus on acute heat stress responses of partners instead of comparing the stable symbiosis starting points, this does help to streamline the paper and focus in on comparisons between stress responses of three different host-symbiont partner combinations. I only have a few more concerns that I would like to see addressed.

Comment 1:

Thank you for providing the table of symbiont count data. As you have remarked, there is a statistical difference in symbiont density between host-symbiont partner combinations: at 25C treatment, CC7-B01 has significantly higher counts than CC7-A4, and, while not significantly different after post-hoc adjustment, CC7-B01 average/median count is still twice as high as H2. And again as remarked in the comment response, in 32C treatment this difference in symbiont abundance between partner combinations weakens.

Since there is evidence that this phenomenon in CC7-B01 is reproducible, I think it is of value to the community to not only cite Herrera 2020, (which briefly provides this information but does not focus on discussing implications) but to explicitly restate this difference in the results and provide some further implications in the discussion. The fact that there are consistently higher stable populations of B01 in CC7 than the homologous symbiont is a critical part of understanding differences between partner combinations, and provides additional information supporting incompatibility of heterologous partners. For example, it could indicate that the heterologous symbiont can repopulate its host above the homologous symbiont levels, which may in itself result in a more severe transcriptomic acute stress response. Also, since statistical comparisons between host-symbiont combinations are made in Figure 3, I think it would be of value to add these statistical comparisons between partner combinations either in text or as another part of the table.

Minor comments

Comment 2:

I appreciate the inclusion of the expr stats which helps allay my concerns about read mapping differences. My only concern now is that others who read this manuscript will not know that these files exist. Perhaps an amendment to the Data accessibility section can include reference to read count data, as I wouldn't call that an "intermediate file".

Comment 3:

I appreciate the clarification regarding Fig 4 and Table S3. I think the phrase in Table S3 "At least one strain had to have the gene expressed to be considered here." led to my confusion. Perhaps changing "expressed" to "differentially expressed above cutoff" would be more clear? As well as including in Table S3 description the clarifying sentences provided in the response: "Fold change values are log2-transformed. Non-zero values represent genes with significant differential expression ($p < 0.05$); while zeroed values represent genes with non-significant differential expression ($p > 0.05$)."

Line comments:

19: remove extra comma

114: fix parenthesis

152: delete "in"

297: add consistent spaces between rep and number in table 1.

386: Post-symbiosis is a bit of a strange term, especially because Aiptasia aposymbiotic comparisons are generated through bleaching and are also technically "post-symbiosis". It would

be good to re-include aposymbiotic reference for clarity e.g. "during symbiosis relative to the aposymbiotic state".

472: period needed.

Reviewer #2 (Remarks to the Author):

I have read the revised manuscript and the author's response to reviews and I feel the manuscript is much improved. All of my previous edits and concerns have been addressed.

Reviewer #3 (Remarks to the Author):

The authors of addressed most of my comments, and the paper is a very nice piece of work, well done.

The addition of symbiont density measurements is welcome (though it should be a figure, rather than a table, with significance testing between symbioses and thermal states). While no differences in symbiont densities are detected due change in temperature, there is a significant doubling of symbiont densities when CC7 is colonised with *B. minutum* rather than *S. microadriaticum* (ANOVA $p = 0.01$, to my fast calculation). Why was the heterologous symbiosis more dense than the heterologous? This is important and should be discussed, if only briefly, as it can affect many of the processes detailed in the manuscript.

188: "Interestingly, CC7 and CC7-B01 appeared to cluster closer under heat stress than under control conditions." Is this well supported? I can't really see it from the PCA plot.

208: "The overall transcriptomic expression patterns were primarily host genotype-driven;" We agree!

224: "By comparing p-values, representing enrichment strength," I would caution against this; GO term analysis is a blunt instrument and comparing fine-scale differences between significant p-values adds little. They are all significant, and this is sufficient. The further analysis of individual genes is appropriate.

237: "exerting higher cellular stress" By what mechanism?

Figure 1 legend: There is too much data interpretation within the legend.

REVIEWERS' COMMENTS:

Reviewer #1 (Remarks to the Author):

I thank the authors for providing revisions to this manuscript which do a good job of clearing up most of my concerns regarding transparency and detail. Though the analysis of the performed experiments are largely restricted to focus on acute heat stress responses of partners instead of comparing the stable symbiosis starting points, this does help to streamline the paper and focus in on comparisons between stress responses of three different host-symbiont partner combinations. I only have a few more concerns that I would like to see addressed.

We thank the reviewer for the favorable assessment of our revisions and for the comments provided. We believe the manuscript is much improved thanks to their helpful input.

Comment 1:

Thank you for providing the table of symbiont count data. As you have remarked, there is a statistical difference in symbiont density between host-symbiont partner combinations: at 25C treatment, CC7-B01 has significantly higher counts than CC7-A4, and, while not significantly different after post-hoc adjustment, CC7-B01 average/median count is still twice as high as H2. And again as remarked in the comment response, in 32C treatment this difference in symbiont abundance between partner combinations weakens.

Since there is evidence that this phenomenon in CC7-B01 is reproducible, I think it is of value to the community to not only cite Herrera 2020, (which briefly provides this information but does not focus on discussing implications) but to explicitly restate this difference in the results and provide some further implications in the discussion. The fact that there are consistently higher stable populations of B01 in CC7 than the homologous symbiont is a critical part of understanding differences between partner combinations, and provides additional information supporting incompatibility of heterologous partners. For example, it could indicate that the heterologous symbiont can repopulate its host above the homologous symbiont levels, which may in itself result in a more severe transcriptomic acute stress response.

We agree with the reviewer, and we have added/changed the following sentences.

In the result section line 297 – 300

Table 1 shows the normalized symbiont density, which revealed no significant differences between the two temperatures. However, we did observe significant differences between CC7 and CC7-B01 at 25 °C, with CC7-B01 exhibiting almost double the normalized symbionts counts ($p < 0.01$).

In the discussion section in lines 447 – 456:

Given that symbiont densities were higher in the heterologous host symbiont combination CC7-B01, the differences in host carbon enrichment were likely due to differences in photosynthate translocation rates. This suggests that reduced metabolic stability and consequent incompatibility already occur under optimal conditions and are further exacerbated under increased temperatures. It is important to restate here that we also observe the same increased symbiont density in CC7-B01 at 25 °C as reported by Rådecker et al.⁷². This increased symbiont density is likely a consequence of the reduced translocation of photosynthates from the symbiont to the host, as it lowers the host's ability to assimilate waste ammonium and its ability to control the availability of nitrogen to the symbionts¹⁴.

Also, since statistical comparisons between host-symbiont combinations are made in Figure 3, I think it would be of value to add these statistical comparisons between partner combinations either in text or as another part of the table.

We thank the reviewer for pointing this out. We now mention these statistical differences in the result section in line 279. The added sentence reads:

However, CC7-B01 also showed significantly higher caspase-3 activation than CC7 ($p < 0.01$) and H2 showed even higher activation than CC7-B01 ($p < 0.05$) during heat stress. These findings corroborated that the *B. minutum* symbiont increases the stress experienced by the host, likely due to increased ROS production⁴⁷.

Minor comments

Comment 2:

I appreciate the inclusion of the expr stats which helps allay my concerns about read mapping differences. My only concern now is that others who read this manuscript will not know that these files exist. Perhaps an amendment to the Data accessibility section can include reference to read count data, as I wouldn't call that an "intermediate file".

Thank you for the suggestion. Added.

Comment 3:

I appreciate the clarification regarding Fig 4 and Table S3. I think the phrase in Table S3 "At least one strain had to have the gene expressed to be considered here." led to my confusion. Perhaps changing "expressed" to "differentially expressed above cutoff" would be more clear? As well as including in Table S3 description the clarifying sentences provided in the response: "Fold change values are log2-transformed. Non-zero values represent genes with significant differential expression ($p < 0.05$); while zeroed values represent genes with non-significant differential expression ($p > 0.05$)."

We have added the two suggested clarifications to the supplementary file in row 2 and 3 of table S3.

Line comments:

19: remove extra comma

Removed.

114: fix parenthesis

Fixed.

152: delete "in"

Deleted.

297: add consistent spaces between rep and number in table 1.

Spaces adjusted.

386: Post-symbiosis is a bit of a strange term, especially because Aiptasia aposymbiotic comparisons are generated through bleaching and are also technically "post-symbiosis". It would be good to re-include aposymbiotic reference for clarity e.g. "during symbiosis relative to the aposymbiotic state".

We have changed the expression accordingly.

472: period needed.

A period has been added to the end of the sentence

Reviewer #2 (Remarks to the Author):

I have read the revised manuscript and the author's response to reviews and I feel the manuscript is much improved. All of my previous edits and concerns have been addressed.

We would like to thank the reviewer for the time and the input provided, both are greatly appreciated.

Reviewer #3 (Remarks to the Author):

The authors of addressed most of my comments, and the paper is a very nice piece of work, well done

We thank the reviewer for this positive assessment of our work and the valuable input that allowed us to improve the manuscript.

The addition of symbiont density measurements is welcome (though it should be a figure, rather than a table, with significance testing between symbioses and thermal states). While no differences in symbiont densities are detected due change in temperature, there is a significant doubling of symbiont densities when CC7 is colonised with *B. minutum* rather than *S. microadriaticum* (ANOVA $p = 0.01$, to my fast calculation). Why was the heterologous symbiosis more dense than the heterologous? This is important and should be discussed, if only briefly, as it can affect many of the processes detailed in the manuscript.

We thank the reviewer for this comment which echoes a similar comment by reviewer 1. We now specifically state these differences in the result section, and we discuss a potential reason in the discussion section.

In the result section line 297 – 300

Table 1 shows the normalized symbiont density, which revealed no significant differences between the two temperatures. However, we did observe significant differences between CC7 and CC7-B01 at 25 °C, with CC7-B01 exhibiting almost double the normalized symbionts counts ($p < 0.01$).

In the discussion section in lines 447 – 456:

Given that symbiont densities were higher in the heterologous host symbiont combination CC7-B01, the differences in host carbon enrichment were likely due to differences in photosynthate translocation rates. This suggests that reduced metabolic stability and consequent incompatibility already occur under optimal conditions and are further exacerbated under increased temperatures. It is important to restate here that we also observe the same increased symbiont density in CC7-B01 at 25 °C as reported by Rådecker et al.⁷². This increased symbiont density is likely a consequence of the reduced translocation of photosynthates from the symbiont to the host, which lowers the host's ability to assimilate waste ammonium and thus increases the availability of nitrogen to the symbionts¹⁴.

188: “Interestingly, CC7 and CC7-B01 appeared to cluster closer under heat stress than under control conditions.” Is this well supported? I can’t really see it from the PCA plot.

While we do report Pearson’s correlation values for CC7-B01 and CC7 at 32 and 25 degrees, we appreciate that the change from a correlation of 0.96 to 0.95 is not significant. As such we have rewritten the sentence to read:

‘CC7 and CC7-B01 appeared to cluster more closely under heat stress and control conditions’

208: “The overall transcriptomic expression patterns were primarily host genotype-driven;”

Thank you!

224: “By comparing p-values, representing enrichment strength,” I would caution against this; GO term analysis is a blunt instrument and comparing fine-scale differences between significant p-values adds little. They are all significant, and this is sufficient. The further analysis of individual genes is appropriate.

We understand the word of caution here and have adjusted the sentence as follows.

‘We observed that CC7 and CC7-B01 had strong enrichment in UPR, while oxidative stress response was most enriched in CC7-B01 and H2.’

237: “exerting higher cellular stress” By what mechanism?

In this sentence we specifically refer to oxidative stress, which we have now highlighted. Sentence now reads:

‘The enriched oxidative stress response in the two strains sharing the same symbionts, CC7-B01 and H2, indicated that the symbiont might be exerting higher oxidative cellular stress on these hosts.’

However, the following sentences are showing that various stress response mechanisms have been triggered, so there is a general stress response of various pathways.

Figure 1 legend: There is too much data interpretation within the legend.

Amended. Now reads:

‘Figure 1 - Principal component analysis of transcriptomic profiles at control and stress conditions. A) Analysis of all three samples transcriptomic profiles under control (25 °C) and heat stress (32 °C). B) Analysis focusing only on CC7 and CC7-B01, to test the separation between control and stress.’